# Wearable intelligent throat enables natural speech in stroke patients with dysarthria

Chenyu Tang [1,13], Shuo Gao [2,13] ✉, Cong Li[2,13], Wentian Yi[1], Yuxuan Jin[3], Xiaoxue Zhai[4], Sixuan Lei[5], Hongbei Meng[2], Zibo Zhang[1], Muzi Xu[1], Shengbo Wang[2], Xuhang Chen[6], Chenxi Wang[2], Hongyun Yang[2], Ningli Wang[7], Wenyu Wang[8], Jin Cao[9], Xiaodong Feng[10], Peter Smielewski[6], Yu Pan[4], Wenhui Song[11], Martin Birchall[12] & Luigi G. Occhipinti[1] ✉

Wearable silent speech systems hold significant potential for restoring communication in patients with speech impairments. However, seamless, coherent speech remains elusive, and clinical efficacy is still unproven. Here, we present an AI-driven intelligent throat (IT) system that integrates throat muscle vibrations and carotid pulse signal sensors with large language model (LLM) processing to enable fluent, emotionally expressive communication. The system utilizes ultrasensitive textile strain sensors to capture high-quality signals from the neck area and supports token-level processing for real-time, continuous speech decoding, enabling seamless, delay-free communication. In tests with five stroke patients with dysarthria, IT's LLM agents intelligently corrected token errors and enriched sentence-level emotional and logical coherence, achieving low error rates (4.2% word error rate, 2.9% sentence error rate) and a 55% increase in user satisfaction. This work establishes a portable, intuitive communication platform for patients with dysarthria with the potential to be applied broadly across different neurological conditions and in multi-language support systems.

Neurological diseases such as stroke, amyotrophic lateral sclerosis (ALS), and Parkinson's disease frequently result in dysarthria—a severe motor-speech disorder that compromises neuromuscular control over the vocal tract. This impairment drastically restricts effective communication, lowers quality of life, substantially impedes the rehabilitation process, and can even lead to severe psychological issues[1-4]. Augmentative and alternative communication (AAC) technologies have been developed to address these challenges, including letter-by-letter spelling systems utilizing head or eye tracking[5-8] and neuroprosthetics powered by brain-computer interface (BCI) devices[9-12]. While head or eye tracking systems are relatively straightforward to implement, they suffer from slow communication speeds. Neuroprosthetics, while transformative for severe paralysis cases, often rely on invasive, complex recordings and processing of neural signals. For individuals retaining partial control over laryngeal or facial muscles, a strong need remains for solutions that are more intuitive and portable (Supplementary Note 1).

A promising solution lies in wearable silent speech devices that capture non-acoustic signals, such as subtle skin vibrations[13-17] or

[1]Department of Engineering, University of Cambridge, Cambridge, UK. [2]School of Instrumentation and Optoelectronic Engineering, Beihang University, Beijing, China. [3]Cavendish Laboratory, University of Cambridge, Cambridge, UK. [4]Department of Rehabilitation Medicine, Beijing Tsinghua Changgung Hospital, Tsinghua University, Beijing, China. [5]Shenzhen International Graduate School, Tsinghua University, Shenzhen, China. [6]Department of Clinical Neurosciences, University of Cambridge, Cambridge, UK. [7]Beijing Tongren Hospital, Capital Medical University, Beijing, China. [8]Thrust of Smart Manufacturing, University of Science and Technology, Guangzhou, China. [9]School of Life Sciences, Beijing University of Chinese Medicine, Beijing, China. [10]Department of Rehabilitation Center, The First Affiliated Hospital of Henan University of Chinese Medicine, Zhengzhou, China. [11]Department of Surgical Biotechnology, University College London, London, United Kingdom. [12]Royal National Ear Nose and Throat and Eastman Dental Hospitals, University College London Hospital, London, UK. [13]These authors contributed equally: Chenyu Tang, Shuo Gao, Cong Li. ✉e-mail: shuo_gao@buaa.edu.cn; lgo23@cam.ac.uk

electrophysiological signals from the speech motor cortex[18-21]. These technologies offer non-invasiveness, comfort, and portability, with potential for seamless daily integration. Yet, despite their promise, current wearable silent speech systems still face three fundamental limitations that hinder their clinical translation and real-world usability. First, most existing studies have been validated primarily on healthy participants, with limited exploration of patient accessibility and adaptability. The resulting gap between laboratory validation and patient-specific deployment prevents these systems from serving individuals with dysarthria or other speech impairments in everyday contexts[13-15]. Second, previous systems often restrict user expression to discrete, word-level decoding within fixed time windows, requiring users to pause and wait before articulating the next word. Such fragmented temporal segmentation disrupts the natural rhythm of silent articulation and makes fluid, continuous communication nearly impossible[13-17]. Third, most approaches rely on a 1:1 mapping between silent articulatory inputs and text outputs. While this direct correspondence works for healthy users, it places excessive physical and cognitive strain on patients, who often experience fatigue even when silently articulating longer sentences (Supplementary Video 1)[13-17]. For these users, a system capable of intelligently expanding shorter or incomplete expressions into coherent, emotionally aligned sentences is crucial for restoring both efficiency and naturalness in communication.

To advance wearable silent speech systems for real-world dysarthria patient use, we developed an AI-driven intelligent throat (IT) system that captures extrinsic laryngeal muscle vibrations and carotid pulse signals, integrating silent speech and emotional states analysis in real-time. The system generates personalized, contextually appropriate sentences that accurately reflect patients' intended meaning (Fig. 1). It employs ultrasensitive textile strain sensors, fabricated using advanced printing techniques, to ensure comfortable, durable, and high-quality signal acquisition[14,22]. By analyzing speech signals at the token level (~100 ms), our approach outperforms traditional time-window methods, enabling continuous, fluent word and sentence expression in real time. Knowledge distillation further reduces computational latency by 76%, significantly enhancing communication fluidity. Large language models (LLMs) serve as intelligent agents, automatically correcting token classification errors and generating

personalized, context-aware speech by integrating emotional states and environmental cues. Pre-trained on a dataset from 10 healthy individuals, the system achieved a word error rate (WER) of 4.2% and a sentence error rate (SER) of 2.9% when fine-tuned on data from five dysarthric stroke patients. Additionally, the integration of emotional states and contextual cues further personalizes and enriches the decoded sentences, resulting in a 55% increase in user satisfaction and enabling dysarthria patients to communicate with fluency and naturalness comparable to that of healthy individuals. Table S1 provides a comprehensive comparison between the IT system and state-of-the-art wearable silent speech systems.

## Results
### The intelligent throat system
The IT system consists primarily of hardware (a smart choker embedding textile strain sensors and a wireless readout printed circuit board (PCB)) and software components (machine learning models and LLM agents). Silent speech signals generated in real time by the user's silent expressions (silently mouthed words in the absence of vocalized sound) are decoded by a token decoding network and synthesized into an initial sentence by the token synthesis agent (TSA). Simultaneously, pulse signals are collected from the smart choker device and processed by an emotion decoding network to determine the user's real-time emotional status. The sentence expansion agent (SEA) intelligently expands the TSA-generated sentence, incorporating personalized emotion labels and objective contextual background data to produce a refined, emotionally expressive, and logically coherent sentence that captures the user's intended meaning (Fig. 1, Supplementary Video 2). Each component of the IT system is elaborated upon in the following sections.

Figure 2a shows the structure of the strain sensing choker screen-printed on an elastic knitted textile (Supplementary Note 3). The choker features two channels located at the front and side of the neck, designed to monitor the strain applied to the skin by the muscles near the throat and the carotid artery (Supplementary Fig. 1). The graphene layer printed on the textile forms ordered cracks along the stress concentration areas of the textile lattice to detect subtle skin vibrations[14]. Silver electrodes are connected to the integrated PCB on the choker. A rigid strain isolation layer with high Young's modulus is

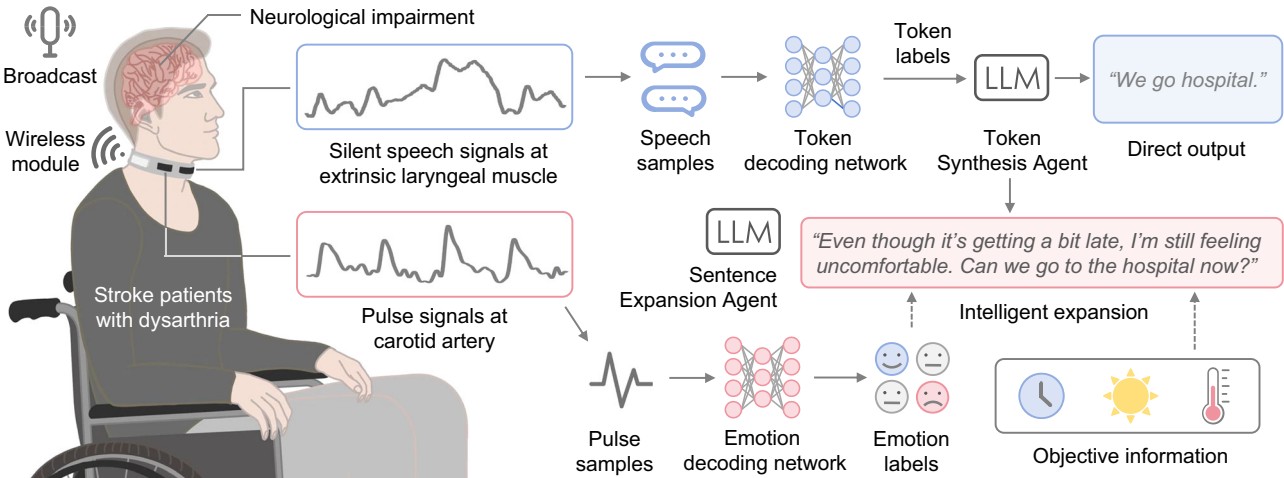

**Fig. 1 | Schematic of the IT developed for stroke patients with dysarthria.** The system captures extrinsic laryngeal muscle vibrations and carotid pulse signals via textile strain sensors and transmits them to the server through a wireless module. Silent speech signals are processed through a token decoding network, which generates token labels for sentence synthesis. Simultaneously, pulse signals are processed by an emotion decoding network to identify emotional states. The system intelligently integrates both emotional states and contextual objective information (e.g., time, environment) to expand the initial decoded sentences. Through a sentence expansion agent, the decoded output is transformed into personalized, fluent, and emotionally expressive sentences, enabling patients to communicate with a fluency and naturalness comparable to healthy individuals. (Note: Due to grammatical differences between Chinese and English, "We go hospital" is a word-for-word translation of the Chinese expression for "Let's go to the hospital").

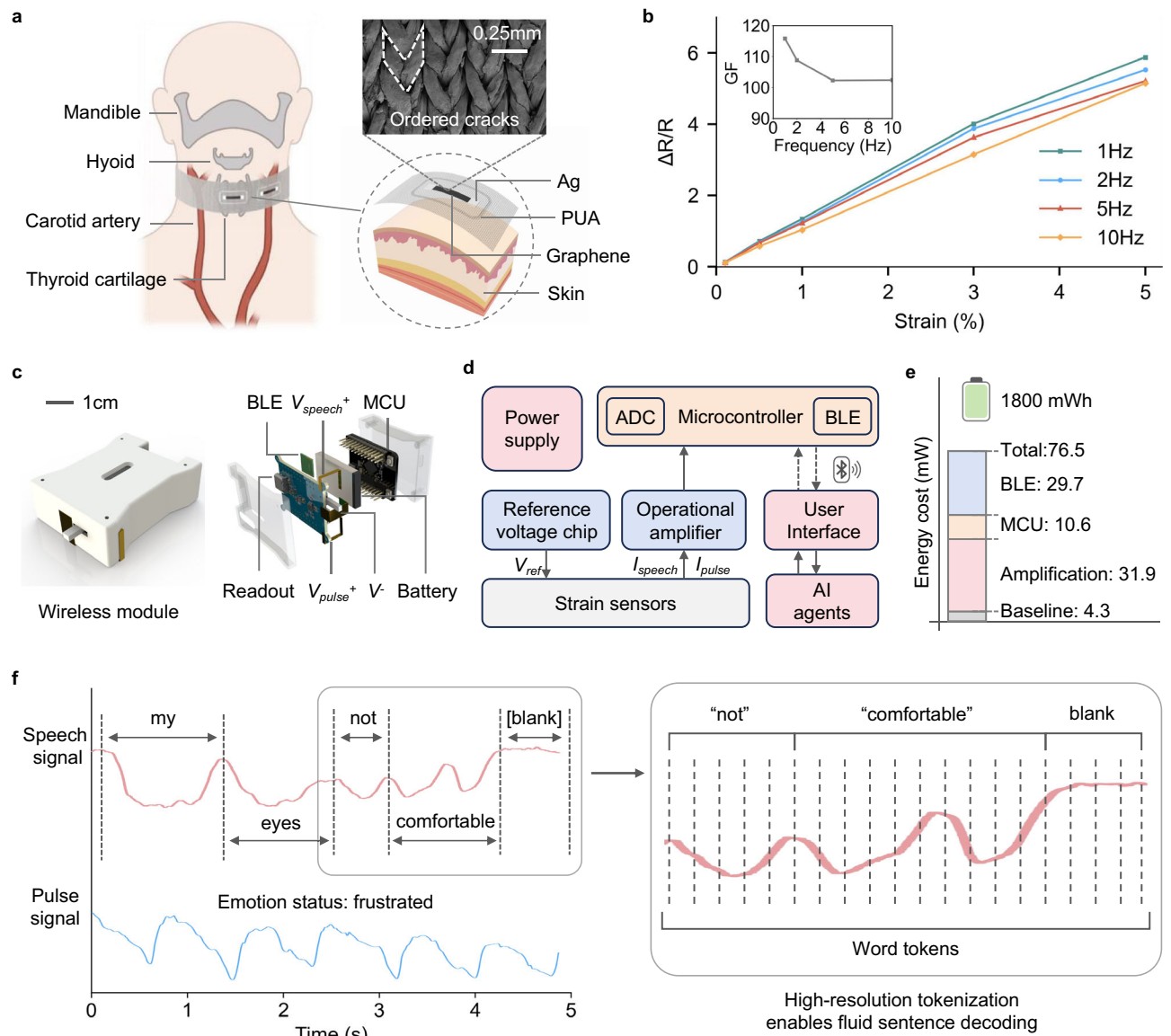

**Fig. 2 | Hardware and data collection of the IT. a** Schematic of a textile-based strain-sensing choker. Two channels are aligned with the carotid artery and center of throat, respectively. Each channel consists of a two-terminal crack-based resistive strain sensor surrounded by a polyurethane acrylate (PUA) stress isolation layer. The top right SEM image shows the spontaneous ordered crack structure of the graphene coating. **b** Relationship between the response to uniaxial stretching (from 0.1% to 5%) and frequency. **c** Exploded view of the internal components of the PCB. **d** Diagram of the system communication. **e** Power consumption of each component during system communication. **f** Schematic of the high-resolution tokenization strategy.

printed around each channel to reduce crosstalk between the two channels and the variable strains caused by wearing. To further validate this effect, we compared devices with and without the isolation layer under identical stretching conditions (Supplementary Fig. 21), confirming that the isolation layer markedly suppresses strain transfer. Due to the difference in Young's modulus between the elastic textile substrate and the strain isolation layer, less than 1% of external strain is transmitted to the interior when wearing the choker, while the internal sensing areas remain soft and elastic (Supplementary Fig. 2)[22]. Furthermore, to quantitatively validate the anisotropic strain sensitivity, we measured the sensor's responses under x-, y-, and z-axis deformation (Supplementary Fig. 20), confirming that the intended x-axis strain dominates the signal while cross-axis interference remains negligible. For uniaxial stretching (x-axis) from 1-10 Hz, the printed textile-based graphene strain sensor shows good linear behavior, producing a response over 10% to subtle strains of 0.1% and maintaining a gauge factor (GF) over 100 during high-frequency stretching

(Fig. 2b), while y- and z-axis deformations contribute negligible signal variations due to the anisotropic crack propagation mechanism. Based on our previous findings and related studies, the 0.1% strain threshold has been validated as sufficient for capturing silent speech-induced muscle vibrations[14,15,17]. Furthermore, our previous studies have confirmed the reliability of the printed textile-based strain sensors with high robustness, durability, and washability, as well as high levels of comfort, biocompatibility, and breathability[14,22].

To operate the system and enable wireless communication between the IT choker and server, the PCB was designed for bi-channel measurements (i.e., silent speech and carotid pulse signals), enabling simultaneous acquisition of speech and emotional cues. The PCB integrates a low-power Bluetooth module (Fig. 2c) for continuous data transmission while optimizing energy efficiency for extended use. Key components of the PCB include an analog-to-digital converter (ADC) for high-fidelity signal digitization and a microcontroller unit (MCU) that manages data processing and transmission (Fig. 2d,

Supplementary Fig. 4, and Supplementary Fig. 5). Power supply, operational amplifiers, and the reference voltage chip are configured to ensure stable signal amplification, catering to the sensitivity requirements of both strain and pulse sensors. For the energy management system, a comprehensive power budget analysis reveals that the designed PCB operates with a total power consumption of 76.5 mW (Fig. 2e). The main power-consuming components are the Bluetooth module (29.7 mW) and amplification circuits (31.9 mW). To extend operational time and support portable use, a 1800 mWh battery was incorporated, providing sufficient capacity for continuous operation throughout an entire day without recharging.

## Token-level speech decoding

Current wearable silent speech systems operate by recognizing discrete words or predefined sentences and lack the ability for continuous, real-time expression analysis typical of the human brain[23]. This limitation arises because these systems rely on fixed time windows (typically 1–3 seconds) for word decoding, requiring users to complete each word within a set interval and pause until the next window to continue[13–21]. Such constraints lead to fragmented expression and unnatural user experience. To address this, we developed a high-resolution tokenization method for signal segmentation (Fig. 2f), dividing speech signals into fine-grained ~100 ms segments for continuous word label recognition. This granular segmentation ensures that each token accurately corresponds to a specific part of a single word and is labeled accordingly. This setup enables users to speak fluidly without worrying about timing constraints, as the system continuously classifies and aggregates tokens into coherent words and sentences. Our optimization determined that a token length of 144 ms offers the ideal balance: it minimizes boundary confusion from longer tokens while avoiding the increased computational demands associated with shorter tokens. This value was empirically determined by gradually reducing token length from 200 ms while monitoring the proportion of boundary-crossing tokens (tokens spanning two adjacent words). A threshold of <5% boundary-crossing tokens was used to define acceptable boundary stability. Shorter tokens were not adopted because the small residual ambiguities they eliminate can already be corrected by the TSA, which applies contextual reasoning and majority voting during word reconstruction. This fine-grained segmentation not only eliminates the unnatural pauses imposed by prior fixed-time-window methods but also ensures that each token retains essential local signal features. Compared to traditional silent speech decoding approaches, which rely on whole-word classification, this token-based approach enables a real-time, continuous speech experience that more closely mimics natural spoken language.

While high-resolution tokenization improves fluidity, shorter tokens inherently contain limited context, making them less effective for accurate word decoding. Temporal machine learning models, like recurrent neural networks (RNN) or transformers, could capture contextual dependencies, but their complexity and computational cost render them suboptimal for wearable silent speech systems[24–26], which prioritize real-time operation. To balance context awareness and computational efficiency, we implemented an explicit context augmentation strategy (Fig. 3a), where each sample consists of N tokens: N-1 preceding tokens provide context, and the current token determines the sample's label. For initial tokens, any missing preceding tokens are padded with blank tokens to ensure completeness. We found $N = 15$ tokens to be optimal (Fig. 3c), with accuracy initially increasing as tokens accumulate, then declining due to insufficient context at lower counts and gradient decay or information loss at higher counts[27]. This strategy enables the use of efficient one-dimensional convolutional neural networks (1D-CNNs) instead of computationally intensive temporal models for token decoding[28,29]. Attention maps reveal that signals from preceding regions indeed

contribute to token decoding, validating the effectiveness of the explicit context augmentation strategy (Supplementary Fig. 10).

To further enhance model efficiency and accuracy on patients' data, we designed the training pipeline shown in Fig. 3b. The model was pre-trained on a larger dataset from healthy individuals and then fine-tuned on the limited patients' data, leveraging shared signal features to enhance patient-specific decoding. After only 25 repetitions per word in few-shot learning, the model achieved a token classification accuracy of 92.2% (Fig. 3d). In contrast, a model trained from scratch using solely patients' data could only reach an accuracy of 79.8%. Additionally, we employed response-based knowledge distillation[30] to transfer knowledge from a larger 1D ResNet-101 model to a smaller 1D ResNet-18, reducing computational load by 75.6% while maintaining high accuracy, with only a 0.9% drop from the teacher model, achieving 91.3% (Fig. 3e). In the inference stage, each segmented token is processed by this trained 1D ResNet-18 model (the final token decoding network) to generate token labels that serve as inputs to the TSA.

Figure 3f, g display the confusion matrix and UMAP feature visualization for token decoding[31]. Over 90% of the classification errors involved confusion between class 0 (blank tokens) and neighboring word tokens. As shown in later analyses of the LLM agent's performance, such boundary errors can be effectively corrected during token-to-word synthesis by the TSA. This knowledge distillation and transfer learning framework ensures that computational efficiency is maximized without sacrificing accuracy. Unlike prior approaches that train models from scratch on small patient datasets, our pipeline generalizes well across individuals, addressing a key challenge in real-world silent speech decoding for dysarthric patients. To further evaluate the discriminability of the IT system on visually and articulatorily similar word pairs, we analyzed five viseme-similar pairs (increase/decrease, ship/sheep, book/look, metal/medal, and dessert/desert). The model achieved an average per-word accuracy of 96.3%, with pairwise confusion rates below 8%, indicating that the system can reliably distinguish between look-alike mouth shapes and subtle articulatory gestures. The detailed confusion matrix is shown in Supplementary Fig. 16. To understand how the system achieves such discriminability, we visualized the raw strain signals and Grad-CAM relevance maps for representative word pairs. As shown in Supplementary Fig. 17, the model consistently focuses on the key articulatory segments where the target words diverge, such as the onset regions in the dessert/desert or book/look pairs. These attention maps confirm that the predictions are driven by meaningful physiological patterns rather than incidental noise or silence segments.

## Decoding of emotional states

To enrich sentence coherence by providing emotional context, we decode emotional states from carotid pulse signals. Emotional changes modulate autonomic nervous activity, which in turn alters the temporal structure of the R-R interval (RRI) within pulse signals, forming measurable physiological representations of affective states[32]. Our machine learning model establishes a direct mapping between these RRI-based temporal representations and corresponding emotional categories. Emotion recognition can be achieved through a range of modalities, including facial expression, audio cues, electromyography, and other physiological signals such as heart rate and blood pressure[33–35]. While multimodal approaches may offer improved accuracy in general populations, they often require additional sensors, power, and computation, limiting system wearability and daily usability. In line with our objective of developing a compact, fully wearable system, we opted for a single-modality strategy centered on carotid pulse signals. This choice reflects a deliberate trade-off between integration and signal diversity. Specifically, stroke patients, our target users, typically exhibit limited mobility, which mitigates motion artifacts and stabilizes pulse dynamics. These

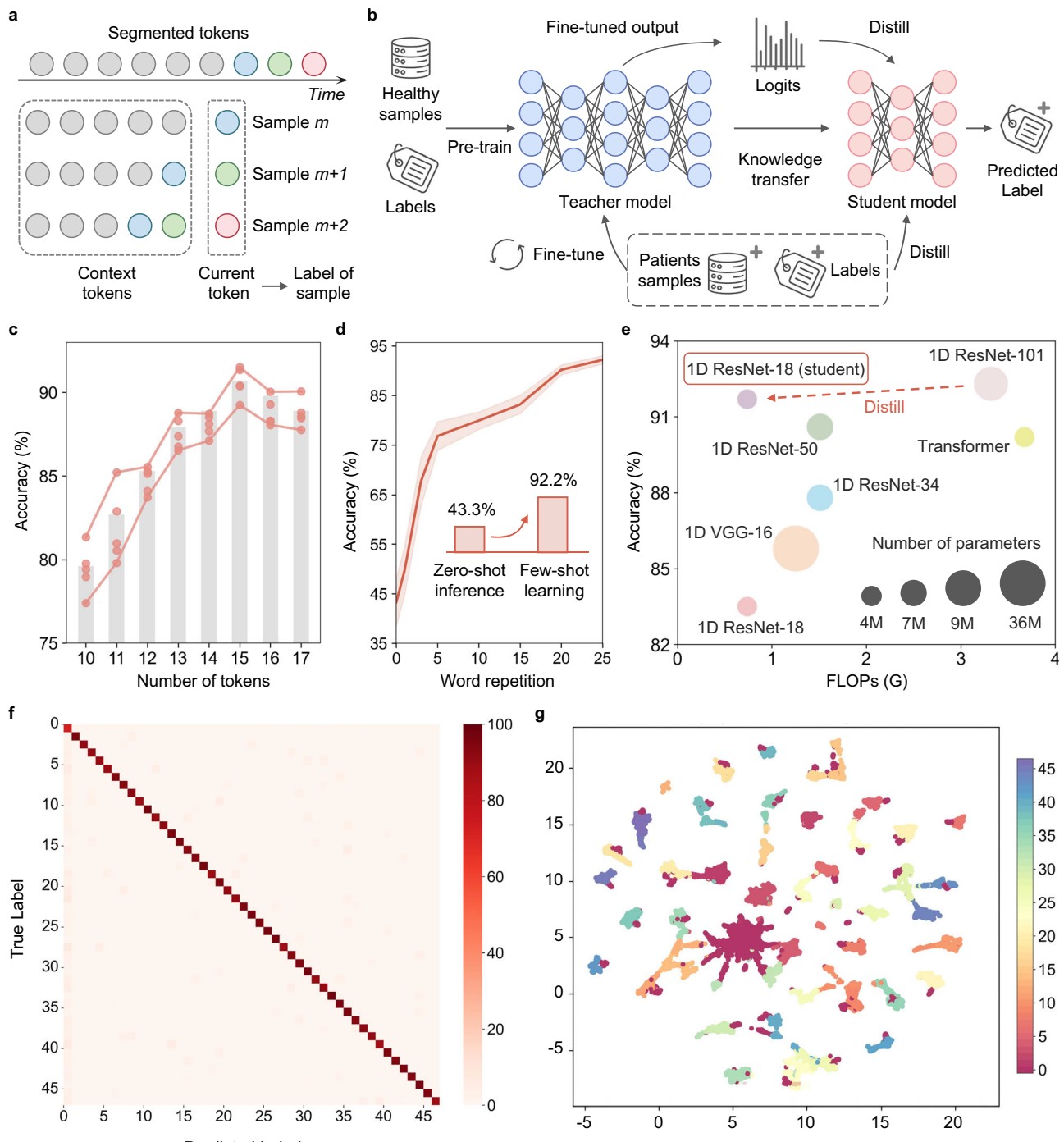

**Fig. 3 | Token-level decoding framework and performance evaluation. a** Explicit context augmentation strategy designed to incorporate contextual information by combining tokens into token samples. **b** Model training pipeline: the teacher model is pre-trained on healthy samples, then fine-tuned on patient samples; knowledge distillation transfers learned features to a student model for efficient prediction. **c** Comparison of decoding accuracy across different numbers of tokens per sample, showing optimal performance when sufficient contextual information is included. **d** Accuracy improvement with word repetition in transfer learning process, demonstrating a jump from zero-shot inference (43.3%) to few-shot learning (92.2%) as repetitions increase. **e** Comparison of model performance across architectures with varying accuracy, FLOPs, and parameter counts; ResNet-101 and ResNet-18 were selected as the teacher and student models, respectively. **f** Confusion matrix for the final student model. **g** UMAP visualization of extracted features from the student model, illustrating token clustering patterns that indicate effective decoding and clear separation of different classes.

conditions allow short-duration pulse segments to provide sufficiently discriminative features for emotion decoding, as demonstrated in our results. Therefore, our use of pulse-based emotion inference is not only aligned with the engineering goals of system simplicity and comfort, but also grounded in the physiological characteristics of the intended clinical population.

Using 5-second windows, we segmented patients' pulse signals into samples to construct a dataset, focusing on three common emotion categories for stroke patients: neutral, relieved, and frustrated (data collection protocol detailed in Methods). Figure 4a shows the discrete Fourier transform (DFT) distributions for each emotion, highlighting distinct frequency characteristics among

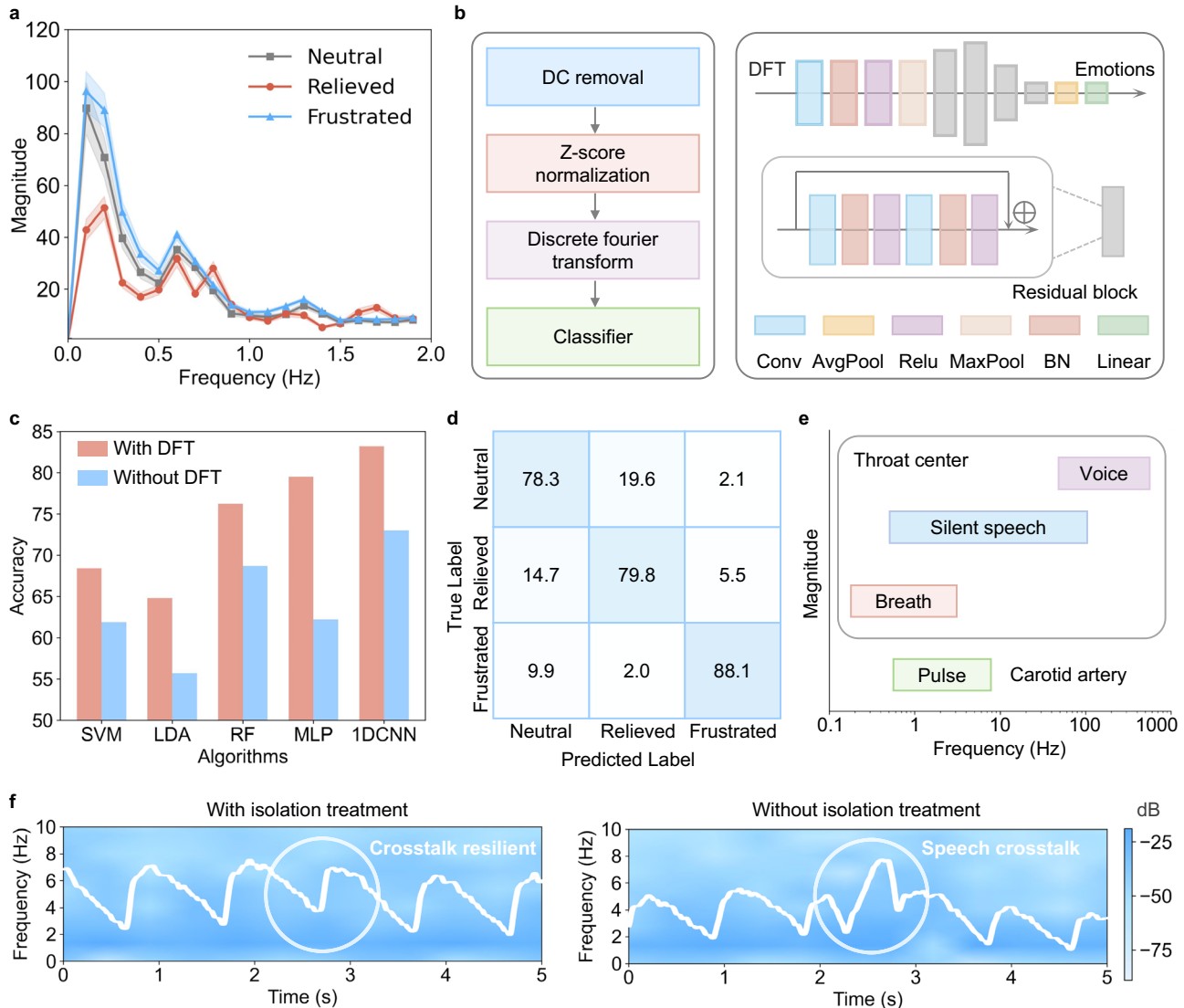

**Fig. 4 | Emotion decoding framework and performance evaluation. a** Frequency domain characteristics of carotid pulse signals across three emotional states (Neutral, Relieved, and Frustrated), showing distinct amplitude patterns. **b** Emotion classification workflow: preprocessing pipeline (left) involving DC removal, Z-score normalization, and discrete Fourier transform (DFT), feeding into a classifier based on a 1DCNN architecture (right) for emotion decoding. **c** Comparison of classification accuracies across machine learning algorithms (SVM, LDA, RF, MLP, and 1DCNN) with and without DFT preprocessing, highlighting improved performance with DFT. **d** Confusion matrix for emotion classification. **e** Frequency and magnitude range of different vibrational signal sources (voice, silent speech, breath, carotid pulse) at neck area. **f** Time-frequency spectrogram of pulse signals with and without strain isolation treatment when vowel "a" both introduced at 2.5 s, demonstrating successful mitigation of speech crosstalk interference after applying the isolation technique.

these emotional states. Accordingly, we incorporated DFT frequency extraction into the decoding pipeline shown in Fig. 4b, where removal of the DC component, Z-score normalization, and DFT are sequentially applied before feeding the values into a classifier for categorization. The DFT-based approach was selected for its ability to represent key characteristics of carotid pulse signals, including power distribution, frequency-domain features, and waveform morphology, within a single transformation. This method enables our end-to-end neural network to automatically extract the most relevant features for emotion classification, eliminating the need for manual feature engineering. Figure 4c illustrates the performance of different classifiers with and without DFT frequency extraction. The results show a significant improvement in decoding accuracy with DFT. The optimal model was the 1D-CNN with DFT, achieving an accuracy of 83.2%, with its confusion matrix displayed in Fig. 4d. The SHAP values reveal that the emotion decoding model primarily focuses on low-frequency signals in the 0-2 Hz range, which is

consistent with the pulse signal range demonstrated by the DFT (Supplementary Fig. 11).

In addition to the silent speech and carotid pulse signals analyzed in this study, various physiological activities generate distinct vibrational signals in the neck area, which can introduce artifacts hindering analysis[36,37]. Figure 4e shows the frequency and magnitude distributions of several prominent signals in this region. Our observations revealed that silent speech exhibits a relatively strong magnitude, and in applications with the IT, vibration can propagate transversely from the throat center to the carotid artery, introducing crosstalk in the pulse signal. Due to the considerable frequency overlap between silent speech and pulse signals, digital filters are non-ideal for effective artifacts suppression[38]. While adding reference channels could theoretically help, it does not align with the goal of a highly integrated IT[39]. To address this issue, we employed a stress isolation treatment using a polyurethane acrylate (PUA) layer, as shown in Fig. 2a, to prevent strain crosstalk propagation along the IT. The theoretical basis of this

isolation strategy has been thoroughly discussed in our previous study[22]. Figure 4f compares pulse signals with and without strain isolation treatment when silent speech occurs concurrently (the vowel "a" introduced at 2.5 s), demonstrating significant crosstalk resilience in the treated IT, with the signal-to-interference ratio improved by more than 20 dB.

### LLM agents for sentence synthesis and intelligent expansion

During clinical observations, we found that stroke patients often experienced marked fatigue even when silently mouthing short phrases, making sustained or complex utterances impractical. To reduce physical effort while preserving the intended message, we incorporated an intelligent expansion option that allows patients to express concise tokens, which are then automatically enriched into complete, contextually appropriate sentences.

To naturally and coherently synthesize sentences that accurately reflect the patient's intended expression from the decoded token and emotion labels, we introduced two LLM agents based on the GPT-4o-mini API (Fig. 5a, Supplementary Note 4): the token synthesis agent (TSA) and the sentence expansion agent (SEA). The TSA merges token labels directly into words silently expressed by the patient and combines them into sentences (left). During this process, it intelligently aggregates consecutive token predictions based on contextual consistency and performs majority-voting reasoning to correct occasional decoding errors or boundary ambiguities from the token decoding network, thereby ensuring accurate word-level reconstruction before sentence formation. The SEA, on the other hand, leverages emotion labels and objective information, such as time and weather, to expand these basic sentences into logically coherent, personalized expressions that better capture the patient's true intent. Through a simple interaction (in this study, two consecutive nods), patients can flexibly choose between direct output and expanded sentences, ensuring that expansion is used only when it aligns with their communication needs.

To optimize the performance of the TSA, we refined the prompt design[40]. First, we optimized the prompt length (Fig. 5b), observing a trend where both WER and SER improved with increasing prompt length up to 400 words before eventually deteriorating for higher lengths. We attribute this trend to the fact that longer prompts provide clearer synthesis instructions, but overly lengthy prompts dilute the model's focus ability. Additionally, we compared performance with and without example cases, where the agent was provided with five examples of token label sequences and their corrected word outputs. Including examples significantly improved synthesis accuracy (Fig. 5c). Finally, we evaluated the effect of providing empirical constraints, which specify typical token counts for words of various lengths. Performance improved considerably when constraints were included (Fig. 5d). Under optimal prompt conditions, TSA achieved its best performance with a WER of 4.2% and an SER of 2.9%.

We also assessed and refined the performance of the SEA. Patient satisfaction with the expanded sentences was evaluated through a questionnaire (see Table S4 for criteria details). Following Chain-of-Thought (CoT) optimization[41] and the inclusion of patient-provided expansion examples, the expanded sentences scored significantly higher across multiple criteria (Fig. 5f). Contribution analysis revealed that emotion labels made a substantial impact on emotion accuracy, while objective information notably improved fluency, jointly contributing to the overall satisfaction with the expanded sentences compared to the basic word-only output (Fig. 5e). Under optimal prompt conditions, the SEA-generated expanded sentences resulted in a 55% increase in overall patient satisfaction compared to the TSA's direct output, raising satisfaction from "somewhat satisfied" to "fully satisfied" levels (Supplementary Fig. 12 and Supplementary Fig. 13).

As shown in Fig. 5f, the core meaning metric remains stable across all sentence expansion conditions. This stability stems from the high accuracy of the token decoding model and TSA, which ensure precise word recognition and correct token synthesis. Since core meaning reflects whether the fundamental subject-verb-object (SVO) structure aligns with the user's intended message, this metric remains largely unchanged after expansion. However, as illustrated in Fig. 5e, additional contextual information—including objective data (e.g., time, weather) and emotion labels—enriches fluency and personalization, significantly improving overall user satisfaction. In both operating modes, sentences generated by the TSA and SEA agents are sent to an open-source text-to-speech model[42], which synthesizes audio that matches the patient's natural voice for playback. In real-world applications, the delay between the completion of the user's silent expression and the sentence playback is approximately 1 second (Supplementary Note 2). This low latency effectively supports seamless and natural communication in practical settings. To assess the long-term adaptability of the IT system, we conducted a follow-up test six months after initial training, observing an increase in WER due to changes in neuromuscular control, which was rapidly restored to initial performance levels after a brief few-shot fine-tuning (five repetitions per words) session (Table S5).

## Discussion

In this work, we introduce the IT, an advanced wearable system designed to empower dysarthric stroke patients to communicate with the fluidity, intuitiveness, and expressiveness of natural speech. Comprehensive analysis and user feedback affirm the IT's high performance in fluency, accuracy, emotional expressiveness, and personalization. This success is rooted in its innovative design: ultrasensitive textile strain sensors capture rich and high-quality vibrational signals from the laryngeal muscles and carotid artery, while high-resolution tokenized segmentation enables users to communicate freely and continuously without expression delays. Additionally, the integration of LLM agents enables intelligent error correction and contextual adaptation, delivering exceptional decoding accuracy (WER < 5%, SER < 3%) and a 55% increase in user satisfaction. While the present study focuses on a defined vocabulary and a small stroke patient cohort, and uses a single-modality approach for emotion decoding, the underlying architecture is designed for scalable adaptation to broader populations, vocabularies, and sensing modalities. The IT thus sets a new benchmark in wearable silent speech systems, offering a naturalistic, user-centered communication aid.

Future efforts in several key areas will guide the continued development of the IT system. First, we are actively expanding our study cohort to include a broader range of dysarthria patients with varying neuromuscular conditions, ensuring that the system is robust across different symptom severities (Table S8). We also plan to recruit participants from diverse linguistic and ethnic backgrounds to capture a wider range of speech patterns and cultural communication norms, enabling multilingual evaluation of the decoding and synthesis pipelines. Second, while the relatively low-intensity activity profiles of stroke patients make pulse-only signals a feasible choice for emotion decoding in the current study—and our controlled trials confirmed stable performance in this setting—future work will incorporate more robust multimodal emotion decoding (e.g., electromyography, respiration, and skin conductance) to improve reliability under diverse real-world conditions, while balancing these gains against constraints on system bulkiness. Third, future hardware iterations will adopt a flexible PCB design to reduce weight and improve conformability to the user's neck. Fourth, we plan to miniaturize the system within an edge-computing framework, enabling fully self-contained, low-latency operation in real-world environments and facilitating seamless daily use. Finally, the current end-of-utterance detection strategy—five consecutive blank tokens—ensures zero-delay operation and aligns with the behavioral tendencies of dysarthric stroke patients, who often prefer short, low-effort expressions. For users who wish to communicate longer or pause-rich utterances, the system can readily

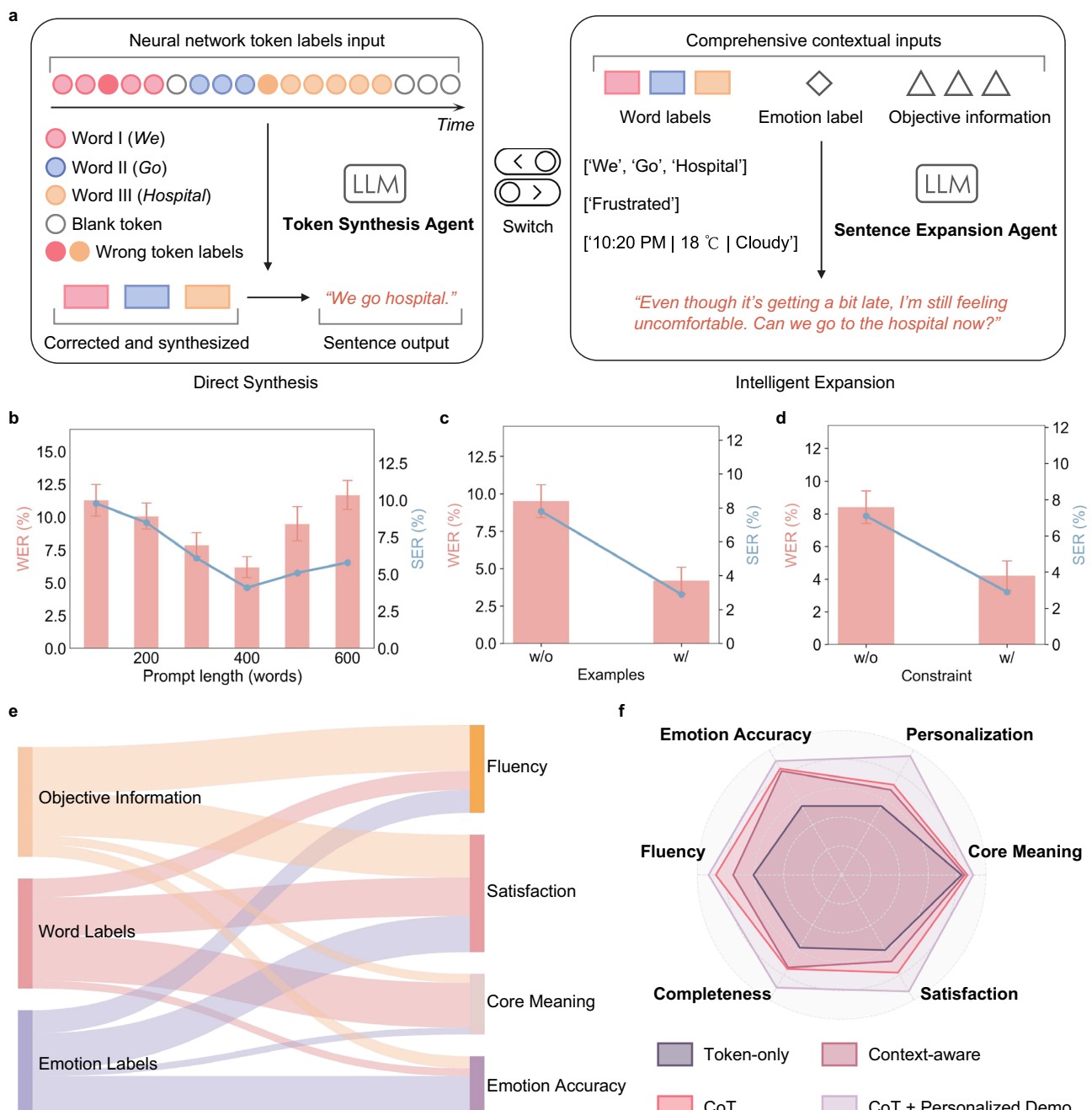

**Fig. 5 | LLM agents framework and performance evaluation. a** Schematic of the IT's LLM agents: Token Synthesis Agent (left) directly synthesizes sentences from neural network token labels, while Sentence Expansion Agent (right) enhances outputs with contextual and emotional inputs. **b** Effect of prompt length on word error rate (WER) and sentence error rate (SER) with optimal performance observed at medium lengths. **c** Influence of example-based few-shot learning on WER and SER, showing a significant reduction when examples are provided. **d** Impact of constrained decoding on WER and SER, demonstrating improved accuracy and sentence structure. **e** Contribution of objective information, word, and emotion labels on key user metrics, including fluency, satisfaction, core meaning, and emotional accuracy (evaluated through ablation experiments). **f** Radar plot comparing performance across various configurations (Token-only, Context-aware, Chain-of-Thought (CoT), and CoT with personalized demonstration) on fluency, personalization, core meaning, satisfaction, completeness, and emotion accuracy. Error bars indicate mean ± s.d.

incorporate an explicit user-controlled cue (e.g., a single nod), which our gesture interface already supports. This modification enables users to indicate expression completion directly, maintaining the system's real-time responsiveness while improving robustness to mid-utterance pauses. Moreover, as the decoding is performed on a fixed sliding window (144 ms per token), the per-token latency and decoding accuracy remain stable regardless of sentence length.

Looking ahead, the advantages of the IT extend beyond enhancing everyday communication; they contribute to the holistic health of neurological patients, encompassing both physical and psychological well-being. The regained fluency in communication allows patients to re-engage in social interactions, reducing isolation and the associated risk of depression. Moreover, effective communication facilitates real-time, personalized adjustments by rehabilitation therapists, supporting patients' recovery from motor impairments like hemiplegia. Future clinical studies will also evaluate the sentence expansion module in larger and more diverse cohorts, incorporating objective performance metrics alongside user-reported satisfaction to ensure practical

relevance and generalizability. Together, these capabilities position the IT as a comprehensive tool for restoring independence and improving quality of life for individuals with neurological conditions.

## Methods

### Materials

TIMREX KS 25 Graphite (particle size of 25 μm) was sourced from IMERYS. Stretchable conductive silver ink was obtained from Dycotec Materials Ltd. Ethyl cellulose was purchased from SIGMA-ALDRICH. Flexible UV Resin Clear was acquired from Photocentric Ltd. The textile substrate, composed of 95% Polyester and 5% spandex, was procured from Jelly Fabrics Ltd.

### Ink formulation

The graphene ink for screen printing was prepared following a reported method. Briefly, 100 g of graphite powder and 2 g of ethyl cellulose (EC) were mixed in 1 L of isopropyl alcohol (IPA) and stirred at 3000 rpm for 30 min. The mixture was then added into a high-pressure homogenizer (PSI-40) at 2000 bar pressure for 50 cycles to obtain graphene dispersion. The graphene dispersion is centrifuged at $5000 \times g$ for 30 min to remove unexfoliated graphite.

### Fabrication of textile strain sensor

The textile substrate was washed with detergent, thoroughly dried, and then treated with UV-ozone for 5 min to clean the surface. Screen printing was performed using a 165 T polyester silk screen on a semi-automatic printer (Kippax & Sons Ltd.) set with a squeegee angle of 45 degrees, a spacer of 2 mm, a coating speed of 10 mm/s, and a printing speed of 40 mm/s. Graphene ink, silver paste, and PUA were successively printed to form the sensing layer, electrodes, and strain isolation layer, respectively. After printing the PUA, the textile was exposed to UV light for 5 minutes. After each printing pass, the textile was air-dried. Following printing, the sensor was dried at 80 °C overnight. A biaxial strain of approximately 10% was then applied to induce the formation of ordered cracks.

Following the standardized fabrication protocol described in our previous work[14,22], our strain sensors exhibit high device-to-device consistency. Across 50 independently fabricated units, the gauge factor (GF) remains consistently centered around ~100, confirming uniform performance. To further verify its applicability across the full signal frequency range involved in pulse (0-10 Hz) and silent-speech (up to ~100 Hz) detection, we evaluated the frequency-dependent strain response of the printed sensor (Supplementary Fig. 19). The GF remained above 98 even at 150 Hz, confirming stable high sensitivity under high-frequency dynamic deformation. Furthermore, long-term stability tests conducted over one month showed no significant degradation in sensor performance, demonstrating the robustness of the sensing mechanism for extended use.

A quantitative head-to-head comparison with MEMS and PVDF sensors under identical conditions further confirmed the superior signal fidelity and comfortability of the proposed textile strain sensor (Table S9). In addition, to evaluate mechanical durability under wearable conditions, printed strips with the same multilayer structure were subjected to 5000 cycles of 0.1–1% tensile loading and to 5000 cycles of bending at ~5 mm radius (Supplementary Fig. 18), both showing stable $\Delta R/R_0$ responses without noticeable degradation.

### Characterization

Scanning Electron Microscopy (SEM) images were taken with a Magellan 400, after sputtering the textile samples with a 5 nm layer of gold to enhance conductivity. Optical images were captured using an Olympus microscope. Tensile properties of the textile strain sensors were evaluated using a Deben Microtest 200 N Tensile Stage and an INSTRON universal testing system. Electrical signals were recorded concurrently with a potentiostat (EmStat4X, PalmSens) and a

multiplexer (MUX, PalmSens). Copper tape was crimped onto the contact pads of the samples, supplemented with a small amount of silver paste to improve electrical contact.

### Wireless PCB for data readout

A custom wireless PCB was developed for efficient, continuous data acquisition and transmission within the IT system. Powered by a TP4065 lithium charger and a 3.3 V regulator, the PCB ensures stable operation via battery or USB. The STM32G431 microcontroller captures silent speech and carotid pulse signals through two ADC channels, with an OPA2192 operational amplifier for high-precision signal conditioning, amplifying low-level signals and enhancing overall data fidelity. A BLE module (BLE-SER-A-ANT) transmits real-time data via UART, enabling seamless, delay-free communication.

To evaluate the comfortability of the wearable system, we conducted a subjective rating assessment with five stroke patients. Each patient rated the system on four key aspects—weight, fit, material flexibility, and long-term wearability—on a 1 to 5 scale (1 = very uncomfortable, 5 = very comfortable). The results are summarized in Table S6, where the overall comfort score was averaged across the four categories. These ratings indicate that the device was generally well-tolerated, with an average overall comfort score of 4.0, though some patients expressed concerns regarding long-term wearability due to rigid PCB components. Future work will focus on further miniaturization and material optimizations to enhance wearability.

To enhance the integration of the IT system and improve overall wearability, a reinforced nylon outer shell was added around the textile choker, providing structural stability without compromising flexibility. The conductive yarns were routed through the gap between the textile choker and the outer casing to connect seamlessly with the PCB. This design modification enhances compactness while ensuring stable signal transmission. The structural improvements and practical wearability of this design are demonstrated in Supplementary Video 3. Future iterations will focus on further miniaturization through the adoption of flexible printed circuit boards (FPCs) and fully integrated wireless communication modules, which will allow direct embedding of the electronics within the textile choker, eliminating the need for external wired connections.

### Assembly and integration of the signal acquisition system

The signal acquisition device consists of a wireless readout module and textile strain sensors. The assembly and wearing configuration are shown in Supplementary Fig. 15. The wireless module was fabricated by enclosing the PCB within a 3D-printed housing for protection and mechanical stability. Electrical contacts on the bottom of the module were extended using copper foil tape to facilitate subsequent connection to the textile sensors.

The textile strain sensors were mounted on an elastic band and connected to the wireless module through conductive yarns stitched laterally along the fabric surface. These conductive yarns pass through the textile layer and are directly linked to the copper foil contacts at the bottom of the wireless device, ensuring reliable electrical connection and mechanical flexibility. This design allows the two sensing channels positioned at the front and side of the neck to interface seamlessly with the wireless PCB positioned on the opposite side, forming a compact, lightweight, and balanced wearable configuration.

### Strain isolation design to control the initial strain set

To account for variability in choker tightness caused by differences in neck circumferences, we implemented a strain isolation strategy similar to the approach used in our previous study[22]. Specifically, a PUA layer with a high Young's modulus was printed around the active sensing area to reduce the influence of tensile deformation introduced by different neck sizes. This layer functions as a strain isolation mechanism, ensuring that the sensor's baseline resistance remains

stable across different users by preventing external mechanical strain from directly affecting the sensing area.

Furthermore, to compensate for any residual variability in tightness after wearing, we integrated a baseline normalization step in the signal processing pipeline. This calibration procedure automatically adjusts for any initial differences in resistance upon device installation, ensuring consistency in signal acquisition. This approach allows the IT system to accommodate different users without requiring manual adjustments, providing robust and reliable data acquisition across varying neck circumferences.

### Separation of silent speech vibrations and carotid pulse signals

To ensure accurate multimodal signal acquisition, two complementary techniques are employed to separate silent speech-induced muscular vibrations and carotid pulse signals. First, a PUA strain isolation layer with a high Young's modulus is printed around the sensing area, effectively preventing mechanical crosstalk between the two channels. Second, the distinct frequency characteristics of these signals are leveraged to apply bandpass filtering: silent speech vibrations are extracted using a 20–200 Hz filter, while carotid pulse signals are processed within a 0.5–5 Hz band. This dual-layer separation approach ensures that each sensor channel accurately captures its respective physiological signal without interference.

### Study design

We recruited 10 healthy subjects (mean age: 25.3 ± 4.1 years; 6 males, 4 females) and 5 stroke patients with dysarthria (mean age: 43 ± 7.8 years; 4 males, 1 female) for silent speech signal collection, in compliance with Ethics Committee approval from the First Affiliated Hospital of Henan University of Chinese Medicine, approval no. 2023HL-142-01. Written informed consent was obtained from all participants for participation and for the publication of de-identified demographic and clinical information.

A corpus was developed consisting of 47 Chinese words commonly used by stroke patients in daily communication and 20 sentences constructed from these words (see Supplementary Tables 2 and 3). The corpus originated from the set of sentences routinely used by our collaborating speech-language therapists during daily rehabilitation sessions with dysarthric patients, representing several hundred therapist-curated daily-communication sentences that collectively contain approximately 350 unique words. To ensure a transparent, unbiased, and reproducible sampling protocol, we performed a computational random selection of 20 sentences from this therapist-provided sentence pool using a fixed random seed (seed = 42). The final 47 words represent the complete set of unique vocabulary items that constitute these 20 selected sentences, forming a minimal but comprehensive vocabulary that captures the core communicative intents encountered in real therapy scenarios. This process can be independently reproduced by applying the same seed to the therapist-curated sentence pool and extracting the unique lexical items that compose the sampled sentences.

Data collection was concluded when subjects completed a predefined number of silent speech repetitions. Specifically, for the healthy subject dataset, we collected 100 repetitions per word and 50 repetitions per sentence. For the patient dataset, we gathered 50 repetitions per word and 50 per sentence. Carotid pulse signals were recorded synchronously with silent speech signals but only for the patient group, ensuring alignment between both modalities.

We did not apply any specific inclusion or exclusion criteria beyond ensuring the technical integrity of the collected signals. The only excluded data were instances where clear sensor readout failures occurred due to connectivity issues. Only signals with evident sensor connection failures (i.e., no observable response) were removed from the dataset. All other signals, including those with motion artifacts or noise, were retained to improve the generalizability of the model to real-world conditions.

### Silent speech data acquisition

All the participants performed silent mouthing, ensuring that no vocalization was present in the collected data. Moreover, prior validation confirms that the device is fully resistant to external sound interference, as evidenced by its unchanged signal response to 100 dB noise exposure[14]. This ensures robust silent speech decoding in diverse acoustic environments.

The healthy subject data serves as a critical baseline for initial model training, enabling the model to establish foundational patterns in silent speech signals. This pre-training facilitates improved generalization and performance when later fine-tuning the model on the limited data from dysarthric patients, ultimately enhancing decoding accuracy and robustness in patient-specific applications. The silent speech signals were segmented into tokens at 144 ms intervals. Each token was combined with the preceding 14 tokens to form a sample, allowing the model to incorporate context. The sample's label corresponds to the word of the current token. The signals were originally recorded at a sampling rate of 10 kHz and subsequently downsampled to 1 kHz before tokenization. Before neural network analysis, each sample was uniformly preprocessed with detrending and z-score normalization.

### Protocol for emotion data collection

Emotional pulse data was collected concurrently with silent speech signals, ensuring synchronized datasets that capture both speech-related and underlying physiological responses. To achieve accurate labeling, each emotion—neutral, relieved, and frustrated—was elicited through a carefully structured protocol involving audio-induced emotional states[43–45]. The emotions were induced via the international affective digitized sounds (2nd Edition; IADS-2)[46]. The selection of the three emotional states (neutral, relieved, and frustrated) was guided by clinical observations and patient feedback. Considering the physiological differences between stroke patients and healthy individuals, emotional pulse signals were collected exclusively from stroke patients. This ensures that the model learns patient-specific pulse-emotion correlations, avoiding confounding effects introduced by healthy individuals' potentially different autonomic and neuromuscular responses. A preliminary survey was conducted with stroke patients and rehabilitation specialists, revealing that these three emotions were the most commonly encountered during communication and therapy sessions. Additionally, these states are distinct enough to be reliably classified using carotid pulse signals. While this study focuses on these core emotions, future iterations of the system aim to incorporate a broader set of emotional states, including anxious, excited, and happy, based on ongoing data collection and annotation efforts. Given that our system is designed specifically for dysarthric stroke patients, training the emotion decoding model exclusively on patient data ensures its clinical relevance and applicability in real-world assistive communication scenarios. This approach prevents potential model drift that could arise from integrating healthy individuals' data, which may not accurately reflect the physiological states of dysarthric patients.

Labeling was verified through collaboration between the participants and the therapist to ensure the successful and reliable induction of each target emotion. To balance sufficient information within each window and achieve the necessary resolution for emotion detection, pulse signals were segmented into 5-second samples. A 50% window overlap was applied to increase the training set size, enhancing model learning and generalization. The signals were originally recorded at a sampling rate of 10 kHz and subsequently downsampled to 200 Hz before analysis.

To evaluate the robustness of the emotion decoding model under natural patient activities, data collection sessions included patients who had just completed hand movement rehabilitation or walking rehabilitation exercises. These activities, commonly performed by stroke patients, were selected to assess whether routine movement-induced variations in carotid pulse signals significantly impact classification accuracy. Although more intense activities (e.g., running) may cause greater distortions in pulse signals, stroke patients typically lack the ability to engage in vigorous exercise. Therefore, our approach remains robust within the stroke patient population.

### Software environment and model training

Signal preprocessing was performed on a MacBook Pro equipped with an M1 Max CPU. Network training was conducted using Python 3.8.13, Miniconda 3, and PyTorch 2.0.1 in a performance-optimized environment. Training acceleration was enabled by CUDA on NVIDIA A100 GPU. The detailed training parameters for all models can be found in Supplementary Fig. 8 and Supplementary Fig. 9. Objective information, including real-time weather conditions, temperature, and time, is retrieved from the local server through a lightweight script. This data is then incorporated into the sentence expansion process to enhance contextual coherence and personalization, without requiring additional sensors or external hardware.

### Reporting summary

Further information on research design is available in the Nature Portfolio Reporting Summary linked to this article.

## Data availability

The datasets supporting this study are publicly available via Zenodo. The code and datasets corresponding to version v1.0.0 of the Intelligent-Throat repository have been deposited at Zenodo under the https://doi.org/10.5281/zenodo.17956161[47].

## Code availability

The code used in this study is publicly available via Zenodo. Version v1.0.0 of the Intelligent-Throat repository, titled "Code and Dataset for 'Wearable intelligent throat enables natural speech in stroke patients with dysarthria'", is archived at Zenodo under the https://doi.org/10.5281/zenodo.17956161[47]. The development repository is hosted on GitHub at: https://github.com/tcy21414/Intelligent-Throat

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

## Acknowledgements

S.G. acknowledges the funding from National Natural Science Foundation of China 62171014 and Beihang Ganwei Project JKF-20240590. L.G.O acknowledges the funding from British Council No. 45371261, UK Engineering and Physical Science Research Council (EPSRC) No. EP/K03099X/1, No. EP/W024284/1, and Haleon through the CAPE partnership contract, University of Cambridge No. G110480.

## Author contributions

C.T., S.G., and L.G.O. conceived and designed the study. C.T., C.L., W.Y., Y.J. and X.Z. developed the methodology. C.T., C.L., W.Y., Y.J., X.Z., S.L., H.M., M.X., C.W., H.Y., W.W., J.C., and X.F. carried out the experiments and data collection. C.T., C.L., W.Y., Y.J. and Z.Z. performed data visualization. S.G. and L.G.O. acquired funding. C.T., S.G., and L.G.O. supervised the project. C.T., C.L., and W.Y. wrote the original draft. C.T., S.G., S.W., X.C., N.W., P.S., Y.P., W.S., M.B., and L.G.O. reviewed and edited the manuscript. All authors commented on the manuscript.

## Competing interests

The authors declare no competing interests.
