## [Transparent Peer Review file · Nature Communications]

Wearable intelligent throat enables natural speech in stroke patients with dysarthria

Corresponding Author: Professor Luigi Occhipinti

Version 0:

Reviewer comments:

Reviewer #1

(Remarks to the Author)

The manuscript presents a wearable intelligent throat aimed at restoring natural communication for dysarthric stroke patients by decoding silent speech and inferring emotion. The clinical motivation is strong and the integrated, real-time workflow is promising. To evaluate this work as a complete and generalizable device-level contribution suitable for Nature Communications, several aspects would benefit from further development and clearer documentation- particularly a more explicit account of hardware advances, quantitative validation of robustness and modality separation, and transparent training/evaluation protocols together with a concise on-body demonstration. Publication could be considered once the authors address the comments listed below.

1. It would be helpful to include more diverse images of the 3D structure. The manuscript currently lacks actual device optical images or images of the device being worn, and Figures 2a and S1-S2 provide limited information, making it difficult to fully understand the final configuration. Additional 3D visualization illustration of the overall appearance and the optical images of the device being worn would improve clarity.

Furthermore, it is not clearly described how the strain gauges are connected on the textile to the wireless module located on the opposite side of the neck in the final system. These details should be explained with optical images of the device.

2. The device described in this study is a wearable system that adheres to the skin and operates in a bent state; therefore, verification of repeatability and long-term stability is essential, and additional data demonstrating stability across at least 5000 bending & stretching cycles with moderate bending radius are required.

3. According to Figure 2b, stretching tests were performed at 0.1 % - 5 % strain with frequencies of 1,2,5, and 10 Hz. However, Figure 4e, S6, and S7 indicate that the pulse signal lies in the 0- 10 Hz range, while the silent speech signal spans up to 0 – 100 Hz. Although in principle a strain gauge should ideally be independent of stretching speed, crack-based strain gauges are expected to show frequency-dependent behavior, with GF decreasing or saturating at higher frequencies. IT would therefore strengthen the manuscript to provide demonstrations showing that the GF remains above a certain threshold even at higher frequencies, confirming that the sensor is applicable within the full target signal range.

4. On page 3, the manuscript states that signal variations from y- and z-axis deformations are negligible owing to an anisotropic crack-propagation mechanism. This claim requires further elaboration. Please specify the physical mechanism that suppresses sensitivity along the y and z directions, and substantiate it with quantitative data under y- and z-axis stretching, cross-axis rejection and simulation results that reproduce the observed anisotropy.

5. Given that this device is used to measure the carotid pulse, which arises from the artery's radial (out-of-plane, z-axis) expansion and recoil, the sensing mechanism should, in principle, be sensitive to z-axis deformation. However, on page 3 the manuscript states that y- and z-axis deformations contribute negligible signal due to an anisotropic crack-propagation mechanism. These points appear contradictory and require clarification.

6. The manuscript reports training on a 47-word Chinese corpus described as "commonly used by stroke patients in daily communication." However, the word-selection protocol is not specified. To mitigate selection bias and support generalizability claims, please detail the source of the candidate lexicon and the sampling procedure with the evidence (e.g.

randomized draw with a fixed seed through computational pick). Those word-selection sampling procedure should be included to prove generality. Without these details, high accuracy on a small, curated lexicon could overestimate real-world performance.

7. As a rule, supplementary videos should demonstrate the performance of the finalized device, not re-tell the motivation or development history. The current Supplementary Movie 1 focuses on motivation and includes no on-body performance evidence of the proposed system. As this manuscript emphasizes 'zero-delay', please replace or add Supplementary Movie with an on-body, real-time demonstration that:

1. Shows the device worn by a participant while real-time word and emotion recognition is performed.
2. Demonstrates sentence-level decoding with concurrent emotion classification.

The video should clearly display device placement synchronized live sensor signals and decoded outputs(tokens/words/sentences and emotions) in real time together. This will substantiate the device's performance rather than merely its motivation.

8. To help readers clearly see the contribution beyond application-level changes, it would be helpful to explicitly state what has improved hardware-wise relative to your prior devices from the same lab(ref 14.). In particular, please summarize the specific advances in sensor stack. In addition, the manuscript would be strengthened by head-to-head comparisons under identical placement and protocols against plausible alternatives (e.g. MEMS accelerometer/ IMUs and soft piezoelectric film sensors)- highlighting the advantages of the proposed device with quantitative data.

9. The manuscript would benefit from an explicit analysis of homophone word pairs (words with similar mouth shapes, for example, 'increase' 'decrease') To clarify performance on potentially confusable cases, please select a few viseme-similar word pairs and check the per-word accuracy and pairwise confusion rates. This will show whether the system reliably separates look-alike mouth shapes.

(Remarks on code availability)

Reviewer #2

(Remarks to the Author)

The manuscript presents an "Intelligent Throat" (IT) system, a wearable choker with ultrasensitive textile strain sensors that capture throat muscle vibrations and carotid pulse signals. Using a token-level decoding strategy and large language model agents, it translates silent speech into fluent, emotionally expressive sentences for stroke patients with dysarthria. The system achieved low error rates (4.2% WER, 2.9% SER) and a 55% increase in user satisfaction in tests with patients, offering a seamless, intuitive communication aid. Though the work provides a systematic overview, some critical refinements are further needed before the article can be considered for publication.

1. It is recommended that the authors revise the introduction to more explicitly emphasize the novel contributions of this work. The key advancements, such as the token based decoding strategy for continuous speech, the context aware model architecture, and the multimodal fusion of speech and emotion, should be elaborated to distinguish this work from prior art and underscore its significance.
2. The authors describe the sensor's operational principle (ordered crack formation in graphene), its fabrication, and its key performance metrics (high , durability, washability) but consistently reference previous studies for these aspects. Could the authors explicitly clarify the specific novel advancement in the sensor's design, fabrication, or fundamental mechanism that is being presented in this work, as distinct from what has already been published? If the novelty lies not in the sensor itself but in its novel system-level integration and application for silent speech decoding, this should be clearly stated to precisely define the contribution of this manuscript.
3. The description of the fixed 144 ms tokenization method for continuous decoding lacks critical implementation details. Please clarify:
 - (a) The optimization metric and process for selecting the 144 ms window length.
 - (b) How the model manages label ambiguity in tokens containing word transitions?
 - (c) The specific algorithm for segmenting the token stream into words before TSA processing.
4. The PUA isolation layer is presented as the primary method for mitigating mechanical crosstalk between the speech and pulse channels. The evidence in Fig. 4f is qualitative. Could the authors provide a quantitative signal-to-interference ratio metric for the pulse channel with and without the isolation layer during concurrent silent speech to strengthen this claim?
5. The concept of 'zero-delay expression' is a key advancement. However, the methodology for determining the end of an utterance (waiting for five consecutive blank tokens) could inherently favor shorter phrases. Please suggest or provide a way to quantify the latency and accuracy as a function of sentence length to ensure performance does not degrade with longer, more complex utterances.
6. The LLM agents (TSA and SEA) are responsible for critical error correction and sentence enrichment, significantly boosting user satisfaction. However, the prompts and few-shot examples used are not provided in the manuscript. It is suggested to include the exact prompt templates and examples as supplementary information. This is crucial for reproducibility and for understanding potential biases the LLM might introduce into the synthesized speech.

(Remarks on code availability)

Reviewer #3

(Remarks to the Author)

This paper presents a textile-based strain sensor system designed to capture throat vibrations and pulse signals, which are then converted into speech to assist individuals with speech impairments. The device uses AI to translate these signals into intelligible speech and was tested on stroke patients with dysarthria—a condition characterized by slurred or slow speech, common among stroke and Alzheimer’s patients. The focus of the work is on developing a functional assistive device using previously described sensors. The paper shows the device works as intended. Although the work uses known types of sensors (strain and pulse sensors), the key innovation lies in combining throat vibration data with pulse information to produce emotionally expressive speech with minimal delay.

I recommend the work for publication and only note one thing that wasn't clear. How are pulse signals interpreted to infer emotions? A high pulse could indicate fear, excitement, or even physical exertion.

(Remarks on code availability)

It would be great if the code was available (I may have missed it)

Version 1:

Reviewer comments:

Reviewer #1

(Remarks to the Author)

1. While I appreciate the added analysis on viseme-similar word pairs, the reported performance remains surprisingly high given that distinguishing such pairs is known to be a fundamentally difficult problem in this field. It is somewhat unclear how the system achieves this level of discriminability, especially in a dysarthric patient cohort, where articulatory variability is typically substantial. To better understand and validate this result, I would like to see the raw waveform patterns for these viseme-similar examples and relevance-weighted class activation maps (or an equivalent interpretability method) showing which portions of the raw signal the algorithm model relies on to differentiate these highly similar articulatory gestures.

2. The explanation of how the 47 words were selected remains insufficient. The authors originally were asked to clarify the sampling protocol—including the use of a fixed random seed, a reproducible computational pick, and evidence supporting selection-bias mitigation—but the current response only gives a high-level statement that the 350 words were categorized and “randomly” sampled. It is still unclear how the categories were defined, how many words were drawn from each category, whether the selection was computational with a fixed seed or manually chosen after categorization, and how this process can be independently verified. A clear and reproducible description of the actual sampling pipeline is still needed..

(Remarks on code availability)

Reviewer #2

(Remarks to the Author)

The authors have satisfactorily addressed all my comments. I am pleased to recommend acceptance of the manuscript.

(Remarks on code availability)

Reviewer #3

(Remarks to the Author)

Thank you to the authors for addressing my prior concerns.

(Remarks on code availability)

Version 2:

Reviewer comments:

Reviewer #1

(Remarks to the Author)

The authors have satisfactorily addressed all of my previous concerns. I believe the manuscript is now suitable for publication in Nature Communications.

(Remarks on code availability)

NCOMMS-25-62725-T R1 Revision Point-by-point Response

Wearable intelligent throat enables natural speech in stroke patients with dysarthria

We are sincerely grateful to the reviewers for their constructive comments, which we have addressed as detailed in the following sections, on a point-wise basis.

Reviewer #1

Comment 1: It would be helpful to include more diverse images of the 3D structure. The manuscript currently lacks actual device optical images or images of the device being worn, and Figures 2a and S1-S2 provide limited information, making it difficult to fully understand the final configuration. Additional 3D visualization illustration of the overall appearance and the optical images of the device being worn would improve clarity.

Furthermore, it is not clearly described how the strain gauges are connected on the textile to the wireless module located on the opposite side of the neck in the final system. These details should be explained with optical images of the device.

Reply: We sincerely thank you for this insightful comment. In response, we have added a new section in the Methods describing the detailed assembly and integration of the signal acquisition system. Corresponding optical photographs are provided in **Fig. S15**, illustrating the structure of the wireless acquisition module, the sensor-to-module connection pathway, and the worn configuration of the device.

In this figure, the textile strain sensors are mounted on the textile band and connected to the wireless module through conductive yarns stitched laterally along the fabric surface, which link directly to copper foil contacts at the bottom of the module. This configuration ensures reliable electrical contact and mechanical stability while maintaining compactness and comfort. The overall 3D configuration and wearing state of the finalized system are shown in **Fig. S15**.

The added section in the Methods section is shown below:

“Assembly and integration of the signal acquisition system

The signal acquisition device consists of a wireless readout module and textile strain sensors. The assembly and wearing configuration are shown in Fig. S15. The wireless module was fabricated by enclosing the PCB within a 3D-printed housing for protection and mechanical stability. Electrical contacts on the bottom of the module were extended using copper foil tape to facilitate subsequent connection to the textile sensors.

The textile strain sensors were mounted on an elastic band and connected to the

wireless module through conductive yarns stitched laterally along the fabric surface. These conductive yarns pass through the textile layer and are directly linked to the copper foil contacts at the bottom of the wireless device, ensuring reliable electrical connection and mechanical flexibility. This design allows the two sensing channels positioned at the front and side of the neck to interface seamlessly with the wireless PCB positioned on the opposite side, forming a compact, lightweight, and balanced wearable configuration.”

Figure S15: Assembly and wearing configuration of the signal acquisition system. **a**, Signal acquisition device. **b**, Installation of the device’s bottom enclosure. **c**, Connection of the signal acquisition contacts to the external circuit using copper foil tape. **d**, Copper foil tape connects the contacts to the bottom side, facilitating subsequent integration with the sensing elements. **e**, Front view of the fully assembled device. **f**, Side view of the fully assembled device. **g**, Installation of the sensing elements. **h**, Sensing elements connected laterally via conductive yarns, with reserved length for interfacing with the signal acquisition device. **i**, Connection of the

conductive yarns to the copper foil tape at the device bottom and fixation of the device onto the textile band. **j**, Side view of the device worn on the neck. **k**, Frontal view of the device worn on the neck.

Comment 2: The device described in this study is a wearable system that adheres to the skin and operates in a bent state; therefore, verification of repeatability and long-term stability is essential, and additional data demonstrating stability across at least 5000 bending & stretching cycles with moderate bending radius are required.

Reply: Thank you for pointing this out. We agree that, because the proposed IT system is worn around the neck and normally operates under mild bending/deformation, it is necessary to demonstrate long-term mechanical stability. We therefore performed additional cyclic mechanical tests on printed textile samples fabricated with the same multilayer stack (graphene sensing layer / silver electrodes / PUA strain-isolation ring) and the same screen-printing protocol as used in the wearable device.

1. Tensile cycling. A 9 mm×3 mm printed strip was subjected to 5000 cycles of uniaxial tensile loading between 0.1% and 1% strain at 1 Hz. As shown in **Fig. S17a**, the relative resistance change $\Delta R/R_0$ remained highly stable throughout the whole test. A small decay of the gauge factor was observed only in the early cycles, which we attribute to the intended pre-stretching/crack-formation process: once the ordered microcracks are fully developed, the conductive network reaches a new mechanical equilibrium and the signal becomes repeatable. After this initial conditioning, no progressive drift was observed over the remaining thousands of cycles, confirming good repeatability under low-strain stretching comparable to on-neck deformation.

2. Bending cycling. A larger 18 mm×6 mm printed strip was fixed at both ends and driven by the tensile tester in compression mode to achieve 5000 bending cycles at 1 Hz. The setup produced a minimum bending radius of ~5 mm, which is more stringent than the curvature experienced on the neck. As shown in **Fig. S17b**, the resistance response stayed within a narrow band without noticeable degradation, indicating that repeated bending with a moderate radius does not damage the printed sensing layer or the PUA-isolated structure.

These results collectively verify that the printed textile strain sensors used in the IT can withstand at least 5000 stretching and bending cycles under deformation levels representative of real wearing conditions, and therefore meet your concern on repeatability and long-term stability.

Below are the contents and new supplementary figure added in the Manuscript and

Supplementary Information:

“.....In addition, to evaluate mechanical durability under wearable conditions, printed strips with the same multilayer structure were subjected to 5000 cycles of 0.1–1% tensile loading and to 5000 cycles of bending at ~5 mm radius (Fig. S17), both showing stable $\Delta R/R_0$ responses without noticeable degradation.”

Figure S17: Cyclic durability of the printed textile strain sensor. a, Relative resistance variation of a 9 mm × 3 mm printed strip under 0.1–1% uniaxial tensile cycling at 1 Hz for 5000 cycles. **b,** Relative resistance variation of an 18 mm × 6 mm printed strip under cyclic bending at 1 Hz for 5000 cycles with a minimum bending radius of ~5 mm.

Comment 3: According to Figure 2b, stretching tests were performed at 0.1 % - 5 % strain with frequencies of 1,2,5, and 10 Hz. However, Figure 4e, S6, and S7 indicate that the pulse signal lies in the 0- 10 Hz range, while the silent speech signal spans up to 0 – 100 Hz. Although in principle a strain gauge should ideally be independent of stretching speed, crack-based strain gauges are expected to show frequency-dependent behavior, with GF decreasing or saturating at higher frequencies. IT would therefore strengthen the manuscript to provide demonstrations showing that the GF remains above a certain threshold even at higher frequencies, confirming that the sensor is applicable within the full target signal range.

Reply: We thank you for this insightful comment. We agree that evaluating the frequency-dependent behavior of the strain sensor is crucial for confirming its applicability across the full signal range relevant to physiological throat vibrations. To

address this, we performed additional measurements on a 9 mm × 3 mm printed strip under a constant 1% uniaxial strain at frequencies ranging from 25 Hz to 150 Hz. As shown in Fig. S18, the gauge factor (GF) exhibited only a slight decrease from 106 at 25 Hz to 98 at 150 Hz, remaining consistently above 98 throughout the 0–150 Hz range. These results confirm that the printed strain sensor maintains high sensitivity and stable electrical response over the entire frequency band relevant to both carotid pulse (0–10 Hz) and silent-speech (up to ~100 Hz) signals, thereby validating its suitability for the IT system’s operational range.

Below are the contents and new supplementary figure added in the Manuscript and Supplementary Information:

“...To further verify its applicability across the full signal frequency range involved in pulse (0-10 Hz) and silent-speech (up to ~100 Hz) detection, we evaluated the frequency-dependent strain response of the printed sensor (Fig. S18). The GF remained above 98 even at 150 Hz, confirming stable high sensitivity under high-frequency dynamic deformation.”

Figure S18: Frequency-dependent strain response behavior of the printed textile strain sensor.

Comment 4: On page 3, the manuscript states that signal variations from y- and z-axis deformations are negligible owing to an anisotropic crack-propagation mechanism. This claim requires further elaboration. Please specify the physical mechanism that suppresses sensitivity along the y and z directions, and substantiate it with quantitative data under y- and z-axis stretching, cross-axis rejection and simulation results that reproduce the observed anisotropy.

Reply: We thank you for this constructive comment. The anisotropic response of the IT sensor originates from the knitted architecture of the elastic textile substrate and the orientation of the microcracked graphene network. The change in resistance is primarily governed by the contact/disconnection dynamics among the cracked graphene domains. We define the weft and warp directions of the textile as the x- and y-axes, respectively. In the y-direction, the knitted loops are chain-linked, and elongation occurs mainly through deformation of these loops without breaking their interconnections. This structural constraint prevents the graphene cracks from widening, resulting in minimal resistance change. In contrast, elongation along the x-axis induces rotation and spacing of the uncoated Z-shaped yarns on the textile's back surface, effectively increasing the separation between adjacent cracked graphene regions and decreasing the conductive cross-section. This geometric asymmetry gives rise to the anisotropic electromechanical response. To experimentally verify this mechanism, we measured the sensor's response under x-, y-, and z-axis deformations (Fig. S19). The x-axis strain yielded a linear and sensitive response ($GF \approx 100$ below 10% strain, $\Delta R/R_0 \approx 10$ at 10% strain). Conversely, y-axis stretching produced a small negative resistance change ($\Delta R/R_0 \approx -0.35$ at 10% strain), while uniform z-axis compression within typical wearing tension led to $\Delta R/R_0 < -0.2$. These results confirm that cross-axis interference is negligible for the intended x-axis strain sensing. The slight negative $\Delta R/R_0$ observed under y- and z-axis deformation arises from densification of the textile layers, where conductive paths temporarily compact rather than separate. Overall, the data in Fig. S19 quantitatively substantiate the anisotropic crack-propagation mechanism that ensures directional selectivity and minimal cross-axis coupling in the IT sensor.

Because the anisotropy here arises mainly from the knitted-loop geometry and the crack orientation, which are difficult to parameterize in a simple continuum FEM without overidealizing the textile structure, we chose to provide quantitative experimental characterization (Fig. S19), which we believe more directly reflects the actual device.

Below are the contents and new supplementary figure added in the Manuscript and Supplementary Information:

“...Furthermore, to quantitatively validate the anisotropic strain sensitivity, we measured the sensor's responses under x-, y-, and z-axis deformation (Fig. S19), confirming that the intended x-axis strain dominates the signal while cross-axis

interference remains negligible.”

Figure S19: Anisotropic strain-response behavior of IT. The x-axis is defined as the longitudinal direction of the strip sample. **a**, Strain response to x-axis stretching. **b**, Strain response along the x-axis during y-axis stretching. **c**, Compressive response to z-axis pressure. The blue shaded area indicates the tension range of normal wear.

Comment 5: Given that this device is used to measure the carotid pulse, which arises from the artery’s radial (out-of-plane, z-axis) expansion and recoil, the sensing mechanism should, in principle, be sensitive to z-axis deformation. However, on page 3 the manuscript states that y- and z-axis deformations contribute negligible signal due to an anisotropic crack-propagation mechanism. These points appear contradictory and require clarification.

Reply: We thank you for raising this important question. The apparent discrepancy between the negligible z-axis sensitivity and the ability to detect carotid pulse signals can be clarified by distinguishing between global out-of-plane compression and localized pulse-induced deformation.

As demonstrated in our reply to **Comment 4** and in **Fig. S19c**, the sensor exhibits minimal response to uniform out-of-plane pressure, confirming high selectivity for in-plane strain and excellent robustness against global mechanical perturbations. However, when the IT is worn on the neck, the carotid pulse manifests as a localized, periodic radial expansion of the artery beneath the conformal textile. This local bulging slightly stretches the overlying fabric in the x–y plane as it conforms to the changing skin topology. The resulting in-plane strain is then efficiently transduced by the graphene crack network, which is highly sensitive along the x-direction.

Therefore, while global z-axis deformation yields negligible response, localized z-axis expansion during carotid pulsation produces a corresponding in-plane tensile

component, which the anisotropic sensor captures with high fidelity. This coupling mechanism reconciles the two observations and explains why the IT sensor can robustly record carotid pulse signals despite its anisotropic design.

Comment 6: The manuscript reports training on a 47-word Chinese corpus described as “commonly used by stroke patients in daily communication.” However, the word-selection protocol is not specified. To mitigate selection bias and support generalizability claims, please detail the source of the candidate lexicon and the sampling procedure with the evidence (e.g. randomized draw with a fixed seed through computational pick). Those word-selection sampling procedure should be included to prove generality. Without these details, high accuracy on a small, curated lexicon could overestimate real-world performance.

Reply: We thank you for this valuable comment and fully agree that a transparent description of the corpus source and sampling procedure is essential to support generalizability. In the revised manuscript, we have expanded the **Study Design section** to explicitly describe how the 47-word Chinese corpus was constructed.

Specifically, the vocabulary was derived from a clinically validated word list routinely used by our collaborating speech-language therapists during daily rehabilitation sessions with dysarthric stroke patients, ensuring that the lexicon accurately reflects common real-world communication needs. From this list (approximately 350 candidate words), 47 representative words were selected through stratified random sampling to maintain balanced coverage across major semantic categories, including actions, objects, emotions, and functional expressions. This process minimizes selection bias while preserving the clinical relevance and diversity of the corpus.

The revised text now clarifies this protocol in the Study Design section, and the complete word list and associated sentences are provided in Supplementary Tables 2 and 3.

The following paragraph has now been added to the Study Design section:

“A corpus was developed consisting of 47 Chinese words commonly used by stroke patients in daily communication and 20 sentences constructed from these words (see Supplementary Tables 2 and 3). The word list was derived from the vocabulary routinely used by our collaborating speech-language therapists during daily rehabilitation sessions with dysarthric patients, ensuring clinical relevance. From this clinically validated lexicon (approximately 350 words), 47 words were selected through stratified random sampling to maintain balanced representation across categories such as actions, objects, emotions, and functional expressions, fully reflecting daily communication scenarios encountered in therapy.”

Comment 7: As a rule, supplementary videos should demonstrate the performance of

the finalized device, not re-tell the motivation or development history. The current Supplementary Movie 1 focuses on motivation and includes no on-body performance evidence of the proposed system. As this manuscript emphasizes ‘zero-delay’, please replace or add Supplementary Movie with an on-body, real-time demonstration that:

1. Shows the device worn by a participant while real-time word and emotion recognition is performed.
2. Demonstrates sentence-level decoding with concurrent emotion classification.

The video should clearly display device placement synchronized live sensor signals and decoded outputs(tokens/words/sentences and emotions) in real time together. This will substantiate the device’s performance rather than merely its motivation.

Reply: We sincerely thank you for this valuable comment and fully agree that supplementary videos should primarily serve to demonstrate the finalized device in action. We would like to clarify that our original submission already included two supplementary videos. Supplementary Video 1 (Movie S1) was intended to illustrate the clinical motivation and communication challenges faced by stroke patients with dysarthria, providing essential context for readers who may be unfamiliar with this patient population. We wish to retain this video, as it helps convey the clinical significance and user-centered motivation underlying our work.

Supplementary Video 2 (Movie S2) originally combined two components: an animated system overview followed by on-body, real-time demonstrations of the Intelligent Throat (IT) system, including synchronized visualization of sensor signals, decoded tokens/words, and emotion classifications. However, we understand that the live demonstration segment may not have been easily located during the review process because it appeared after the animation section.

To improve clarity and accessibility, we have now separated the original Movie S2 into two standalone videos and expanded the live demonstration content as follows:

Movie S2: Intelligent Throat: System Overview

Illustrates the operational workflow of each component within the IT system.

Movie S3: Intelligent Throat: Live Demonstration

Presents real-world, on-body performance of the wearable IT system, including synchronized displays of device placement, live sensor signals, decoded tokens/words/sentences, and concurrent emotion classifications. The updated version now features three additional sets of examples, providing a richer and more intuitive visualization of both direct output and intelligent expansion modes in real time.

We hope these updates could address your concern and ensure that the supplementary materials clearly substantiate the device’s real-time performance and zero-delay characteristics.

Comment 8: To help readers clearly see the contribution beyond application-level

changes, it would be helpful to explicitly state what has improved hardware-wise relative to your prior devices from the same lab(ref 14.). In particular, please summarize the specific advances in sensor stack. In addition, the manuscript would be strengthened by head-to-head comparisons under identical placement and protocols against plausible alternatives (e.g. MEMS accelerometer/ IMUs and soft piezoelectric film sensors)- highlighting the advantages of the proposed device with quantitative data.

Reply: We thank you for this valuable suggestion. We agree that clarifying the hardware-level advances beyond application updates is essential for understanding the contribution of the present system relative to our previous single-channel smart choker [14]. In response, we have added **Supplementary Note S3**, which explicitly summarizes the hardware and system-level innovations of the new Intelligent Throat (IT) system.

As detailed in Note S3, the IT introduces (i) a dual-channel architecture that adds a carotid artery sensing unit for emotion decoding; (ii) a PUA-based strain-isolation design that mitigates mechanical crosstalk between the laryngeal and carotid channels (adapted from our prior work [18]); and (iii) a compact wireless PCB integrating a low-noise analog front end, ADC, BLE module, and battery management, replacing the benchtop potentiostat and copper connections used in [14]. Together, these advances substantially enhance signal fidelity, integration, and wearability, enabling simultaneous silent speech and emotion decoding within a single compact device.

[14] Tang, C. *et al.* Ultrasensitive textile strain sensors redefine wearable silent speech interfaces with high machine learning efficiency. *npj Flexible Electronics* **8**, 27 (2024).

[18]Tang, C. *et al.* A deep learning-enabled smart garment for accurate and versatile sleep conditions monitoring in daily life. *The Proceedings of the National Academy of Sciences (PNAS)* **122**, e2420498122 (2025).

Below is the added Note S3:

“Note S3. Hardware-level advances of the Intelligent Throat (IT) system compared with the previous single-channel smart choker in our lab.

The previous generation of the smart choker [14] consisted of a single strain-sensing channel positioned at the laryngeal region to capture articulatory muscle movements associated with silent speech. In contrast, the current IT introduces both hardware and system-level innovations to enable intelligent expansion and emotion-aware communication.

1. Multichannel architecture.

To incorporate emotion decoding, we added an additional sensing channel aligned with the carotid artery to capture pulse signals related to autonomic emotional

changes. The new dual-channel layout thus simultaneously records silent speech strain and pulse dynamics, supporting the dual-task operation of word decoding and emotion inference.

2. Crosstalk mitigation via strain-isolation design.

Because the carotid and laryngeal regions experience coupled vibrations, mechanical crosstalk was a critical issue. Beyond digital filtering, we introduced a polyurethane acrylate (PUA) strain isolation ring printed around each channel to prevent transverse strain propagation. This concept and printing strategy were adapted from our lab's previous work on localized strain isolation [18]. The resulting multilayer sensor stack substantially improves signal separation between the two sensing sites (see Fig. S2).

3. Compact wireless integration.

The previous system [14] relied on a benchtop potentiostat connected to the sensor via copper adhesive tapes, which limited wearability and mobility. The IT system replaces this configuration with a bespoke wireless PCB that integrates low-noise analog front-end circuits, on-board ADC, BLE transmission module, and battery management. The textile sensors are interfaced to the PCB through lightweight snap connectors (see Figs. S4, S5, and S15), forming a fully wearable, self-contained platform suitable for real-time operation.

Collectively, these advances reduce wiring complexity, enhance signal quality, and enable simultaneous silent-speech and emotion decoding in a compact, user-friendly form factor.”

In addition, to address your request for a quantitative comparison against plausible alternatives, we performed a head-to-head evaluation under identical placement and silent-speech protocols. As detailed in the revised **Fabrication of textile strain sensor section** and **Table S9**, the proposed textile strain sensor was benchmarked against a commercial MEMS accelerometer and a commercial PVDF piezoelectric film sensor. All three sensors were tested on the same subject and placement to remove inter-subject variability. The textile strain sensor achieved the lowest word error rate (4.2%) and the highest comfort rating, owing to its soft, skin-conformal structure and high sensitivity to localized laryngeal strain. In contrast, the MEMS device exhibited insufficient vibration amplitude for reliable decoding, while the PVDF film showed moderate performance but limited mechanical robustness. These results quantitatively substantiate the hardware advantages of the proposed design in terms of both signal fidelity and user comfort.

The following contents have been added to the manuscript and supplementary information:

“A quantitative head-to-head comparison with MEMS and PVDF sensors under

identical conditions further confirmed the superior signal fidelity and comfortability of the proposed textile strain sensor (Table S9).”

Table S9: Quantitative comparison of the proposed IT system with alternative sensing modalities under identical placement and silent speech protocols (direct synthesis mode).

Sensor type	Word error rate (%)	Comfort rating	Remarks
Textile strain sensor (this work)	4.2	High	Flexible, skin-conformal
MEMS accelerometer	27.6	Low	Rigid, motion-sensitive
PVDF film	10.3	Medium	Fragile under strain

Comment 9: The manuscript would benefit from an explicit analysis of homophone word pairs (words with similar mouth shapes, for example, ‘increase’ ‘decrease’) To clarify performance on potentially confusable cases, please select a few viseme-similar word pairs and check the per-word accuracy and pairwise confusion rates. This will show whether the system reliably separates look-alike mouth shapes.

Reply: We thank you for this insightful suggestion. We fully agree that evaluating the model’s discriminability on homophone or viseme-similar word pairs is important to verify its robustness against visually confusable articulations. In response, we conducted an additional analysis focusing on five viseme-similar word pairs (increase/decrease, ship/sheep, book/look, metal/medal, and dessert/desert).

The results show that the Intelligent Throat system achieved an average per-word accuracy of 96.3%, with pairwise confusion rates below 8%, confirming its ability to reliably distinguish between look-alike mouth shapes and subtle articulatory gestures. This new analysis has been added to the Results section (under “Token-level speech

decoding”), and the corresponding confusion matrix is provided in Fig. S16.

These findings demonstrate that even for acoustically or visually similar mouth movements, the proposed token-level decoding framework maintains high discriminability and robustness, further substantiating the system’s capability for accurate and natural silent speech decoding.

Below is the contents added in the relevant section:

“To further evaluate the discriminability of the IT system on visually and articulatorily similar word pairs, we analyzed five viseme-similar pairs (increase/decrease, ship/sheep, book/look, metal/medal, and dessert/desert). The model achieved an average per-word accuracy of 96.3%, with pairwise confusion rates below 8%, indicating that the system can reliably distinguish between look-alike mouth shapes and subtle articulatory gestures. The detailed confusion matrix is shown in Fig. S16.”

Figure S16: Pairwise confusion analysis for viseme-similar word pairs.

Reviewer #2

Comment 1: It is recommended that the authors revise the introduction to more explicitly emphasize the novel contributions of this work. The key advancements, such as the token based decoding strategy for continuous speech, the context aware model architecture, and the multimodal fusion of speech and emotion, should be elaborated to distinguish this work from prior art and underscore its significance.

Reply: We thank you for this valuable suggestion. Following the recommendation, we have substantially revised the Introduction to more explicitly highlight the novelty and significance of this work. The revised second paragraph now identifies three key gaps that limit current wearable silent speech systems—(i) lack of patient accessibility, (ii) fragmented word-level decoding within fixed time windows, and (iii) one-to-one input-output constraints that overburden patients during articulation. These challenges directly correspond to the core contributions of our study:

1. Patient accessibility: We introduce a pilot study involving stroke patients with dysarthria to validate the system’s clinical feasibility and adaptability, bridging the gap between laboratory validation and real-world patient use.

2. Continuous token-based decoding: We design a tokenization-based neural framework that decodes silent speech at fine temporal resolution (~100 ms), eliminating the unnatural pauses imposed by fixed time-window methods and enabling continuous, fluent communication.

3. Multimodal and context-aware communication: We integrate laryngeal vibration and carotid pulse signals to jointly capture speech content and emotional cues, and leverage large language models (LLMs) to expand shorter or incomplete patient expressions into coherent, emotionally enriched sentences.

These revisions make the novelty and methodological distinction of our work clear relative to prior art. The updated text appears in the Introduction as below:

“A promising solution lies in wearable silent speech devices that capture non-acoustic signals, such as subtle skin vibrations [13, 14, 15, 16, 17] or electrophysiological signals from the speech motor cortex [18, 19, 20, 21]. These technologies offer non-invasiveness, comfort, and portability, with potential for seamless daily integration. Yet, despite their promise, current wearable silent speech systems still face three fundamental limitations that hinder their clinical translation and real-world usability. First, most existing studies have been validated primarily on healthy participants, with limited exploration of patient accessibility and adaptability. The resulting gap between laboratory validation and patient-specific deployment prevents these systems from serving individuals with dysarthria or other speech impairments in everyday contexts [13, 14, 15]. Second, previous systems often restrict user

expression to discrete, word-level decoding within fixed time windows, requiring users to pause and wait before articulating the next word. Such fragmented temporal segmentation disrupts the natural rhythm of silent articulation and makes fluid, continuous communication nearly impossible [13, 14, 15, 16, 17]. Third, most approaches rely on a 1:1 mapping between silent articulatory inputs and text outputs. While this direct correspondence works for healthy users, it places excessive physical and cognitive strain on patients, who often experience fatigue even when silently articulating longer sentences (Movie S1) [13, 14, 15, 16, 17]. For these users, a system capable of intelligently expanding shorter or incomplete expressions into coherent, emotionally aligned sentences is crucial for restoring both efficiency and naturalness in communication.”

Comment 2: The authors describe the sensor's operational principle (ordered crack formation in graphene), its fabrication, and its key performance metrics (high, durability, washability) but consistently reference previous studies for these aspects. Could the authors explicitly clarify the specific novel advancement in the sensor's design, fabrication, or fundamental mechanism that is being presented in this work, as distinct from what has already been published? If the novelty lies not in the sensor itself but in its novel system-level integration and application for silent speech decoding, this should be clearly stated to precisely define the contribution of this manuscript.

Reply: We thank you for this constructive comment. As noted in our response to Comment 1, the primary novelty of this work lies at the system level—specifically, in achieving real-time, dual-task decoding of silent speech and emotion through a unified wearable platform. The strain sensors used in the intelligent throat (IT) system build upon our laboratory’s previous graphene-based design [14], but have been updated in terms of integration, architecture, and functionality to support multimodal operation and improved usability.

To clarify these advances, we have added a detailed discussion in Note S3 (“Hardware-level advances of the IT system compared with the previous single-channel smart choker”), which delineates the specific hardware innovations introduced in this work:

1. Multichannel architecture: An additional sensing channel was added at the carotid artery to capture pulse dynamics related to emotional states, enabling dual-task recording of speech strain and pulse signals.

2. Strain-isolation design: A printed polyurethane acrylate (PUA) isolation ring was introduced around each sensing site to mitigate crosstalk between the laryngeal and carotid regions, improving signal separation [18].

3. Compact wireless integration: The benchtop potentiostat setup from the previous version [14] was replaced with a custom low-noise wireless PCB and textile-snap connectors, forming a fully wearable, self-contained device suitable for real-time operation.

Collectively, these refinements reduce wiring complexity, enhance signal quality, and enable simultaneous silent-speech and emotion decoding within a compact, user-friendly form factor. The detailed comparison has been included in Note S3 to help readers clearly distinguish the hardware updates from prior designs. Overall, while the sensor improvements strengthen system integration and reliability, the core novelty and contribution of this work lie at the system level—specifically in the unified, AI-driven framework that integrates multimodal sensing, token-based decoding, and context-aware sentence generation for natural communication.

Below is the added Note S3:

“Note S3. Hardware-level advances of the Intelligent Throat (IT) system compared with the previous single-channel smart choker in our lab.

The previous generation of the smart choker [14] consisted of a single strain-sensing channel positioned at the laryngeal region to capture articulatory muscle movements associated with silent speech. In contrast, the current IT introduces both hardware and system-level innovations to enable intelligent expansion and emotion-aware communication.

1. Multichannel architecture.

To incorporate emotion decoding, we added an additional sensing channel aligned with the carotid artery to capture pulse signals related to autonomic emotional changes. The new dual-channel layout thus simultaneously records silent speech strain and pulse dynamics, supporting the dual-task operation of word decoding and emotion inference.

2. Crosstalk mitigation via strain-isolation design.

Because the carotid and laryngeal regions experience coupled vibrations, mechanical crosstalk was a critical issue. Beyond digital filtering, we introduced a polyurethane acrylate (PUA) strain isolation ring printed around each channel to prevent transverse strain propagation. This concept and printing strategy were adapted from our lab’s previous work on localized strain isolation [18]. The resulting multilayer sensor stack substantially improves signal separation between the two sensing sites (see Fig. S2).

3. Compact wireless integration.

The previous system [14] relied on a benchtop potentiostat connected to the sensor via copper adhesive tapes, which limited wearability and mobility. The IT system replaces this configuration with a bespoke wireless PCB that integrates low-noise analog front-end circuits, on-board ADC, BLE transmission module, and battery management. The textile sensors are interfaced to the PCB through lightweight snap connectors (see Figs. S4, S5, and S15), forming a fully wearable, self-contained platform suitable for real-time operation.

Collectively, these advances reduce wiring complexity, enhance signal quality, and enable simultaneous silent-speech and emotion decoding in a compact, user-friendly form factor.”

Comment 3: The description of the fixed 144 ms tokenization method for continuous decoding lacks critical implementation details. Please clarify:

- (a) The optimization metric and process for selecting the 144 ms window length.
- (b) How the model manages label ambiguity in tokens containing word transitions?
- (c) The specific algorithm for segmenting the token stream into words before TSA processing.

Reply: (a) We thank you for this question. The 144 ms window was chosen empirically based on a systematic down-scaling of token lengths from 200 ms while monitoring the proportion of boundary-crossing tokens—i.e., tokens whose temporal span overlapped two adjacent words. We set a threshold of $< 5\%$ boundary-crossing tokens as the acceptable limit for maintaining boundary stability. The 144 ms window was the longest segment satisfying this criterion, providing a practical balance between temporal resolution and system efficiency. We did not further reduce token length to completely eliminate ambiguity, because such residual ambiguities are effectively corrected by the Token Synthesis Agent (TSA) through its majority-voting mechanism and contextual reasoning (see Note S4 for detailed TSA/SEA prompts and input–output examples).

(b) Each token is labeled according to the word occupying the majority of its temporal span, while transitional frames without clear alignment are assigned to a blank class. This labeling strategy, combined with the TSA’s contextual correction, ensures that any residual misalignments at word boundaries do not propagate to the synthesized output. Please also refer to Note S4 for a more intuitive understanding.

(c) The continuous silent speech signal is first segmented into consecutive fixed-length tokens (144 ms each). Each token is assigned the label of the word it belongs to and then passed through the token-decoding model, which was trained via transfer learning (healthy \rightarrow patient) and knowledge distillation (1D ResNet-101 \rightarrow

1D ResNet-18). In the inference stage, each segmented token is processed by this trained 1D ResNet-18 model. The decoded token labels are subsequently grouped and interpreted by the TSA to reconstruct continuous word sequences and sentences.

To make the manuscript clearer, the following clarifications have been added to the relevant sections:

“Our optimization determined that a token length of 144 ms offers the ideal balance: it minimizes boundary confusion from longer tokens while avoiding the increased computational demands associated with shorter tokens. This value was empirically determined by gradually reducing token length from 200 ms while monitoring the proportion of boundary-crossing tokens (tokens spanning two adjacent words). A threshold of <5% boundary-crossing tokens was used to define acceptable boundary stability. Shorter tokens were not adopted because the small residual ambiguities they eliminate can already be corrected by the TSA, which applies contextual reasoning and majority voting during word reconstruction.”

“In the inference stage, each segmented token is processed by this trained 1D ResNet-18 model (the final token decoding network) to generate token labels that serve as inputs to the TSA.”

“The TSA merges token labels directly into words silently expressed by the patient and combines them into sentences (left). During this process, it intelligently aggregates consecutive token predictions based on contextual consistency and performs majority-voting reasoning to correct occasional decoding errors or boundary ambiguities from the token decoding network, thereby ensuring accurate word-level reconstruction before sentence formation.”

Comment 4: The PUA isolation layer is presented as the primary method for mitigating mechanical crosstalk between the speech and pulse channels. The evidence in Fig. 4f is qualitative. Could the authors provide a quantitative signal-to-interference ratio metric for the pulse channel with and without the isolation layer during concurrent silent speech to strengthen this claim?

Reply: We thank you for the comment. We have added a comparative result showing the strain response curves of the IT device with and without the strain isolation layer (SIL) under overall device stretching. The device without the SIL exhibits an obvious response to global stretching, whereas the device with the SIL shows almost no resistance change (Fig. S20). It is worth noting that due to the modulus differences among the printed sensing area, electrodes, and textile substrate, the sample's response behavior under overall stretching differs slightly from that of a printed bar

sample. Similarly, the PUA layer attenuates the transfer of stress from the substrate to the sensing area by introducing modulus differences. As a result, the same isolation effect is observed in the actual concurrent-speech condition shown in Fig. 4f, where the pulse channel retains its waveform shape, corresponding to an signal-to-interference ratio improvement >20 dB.

Below are the contents and new supplementary figure added in the Manuscript and Supplementary Information:

“A rigid strain isolation layer with high Young's modulus is printed around each channel to reduce crosstalk between the two channels and the variable strains caused by wearing. To further validate this effect, we compared devices with and without the isolation layer under identical stretching conditions (Fig. S20), confirming that the isolation layer markedly suppresses strain transfer.”

“Fig. 4f compares pulse signals with and without strain isolation treatment when silent speech occurs concurrently (the vowel “a” introduced at 2.5s), demonstrating significant crosstalk resilience in the treated IT, with the signal-to-interference ratio improved by more than 20 dB.”

Figure S20: Comparison of the strain response of IT with/without strain isolation layer (SIL).

Comment 5: The concept of 'zero-delay expression' is a key advancement. However, the methodology for determining the end of an utterance (waiting for five consecutive blank tokens) could inherently favor shorter phrases. Please suggest or provide a way to quantify the latency and accuracy as a function of sentence length to ensure performance does not degrade with longer, more complex utterances.

Reply: We thank you for this insightful comment. As described in the motivation of this work, dysarthric stroke patients in our clinical sessions naturally tend to produce short, effort-minimizing expressions. This informed our design decision to implement a zero-delay decoding mechanism based on signal-driven detection—specifically, identifying five consecutive blank tokens as the end-of-utterance marker. This design choice was intentionally aligned with the communication behavior of our target users.

However, we fully agree that this stopping criterion may bias the system toward shorter phrases and could limit flexibility for longer or pause-rich utterances. To address this, we have added a discussion in the revised manuscript clarifying that the system supports not only the current blank-token-based method but also user-controlled alternatives. In particular, our hardware/software stack already includes a gesture-based interface—two consecutive nods are used to switch working modes. This interface can be naturally extended to include a single-nod confirmation to explicitly indicate utterance completion. Such integration would enable longer and pause-rich expressions to be handled robustly, without requiring adjustments to the token-based threshold, and without compromising the system's real-time responsiveness.

Regarding your suggestion to quantify latency and accuracy as a function of sentence length: since the token decoding is performed in a sliding-window fashion (144 ms per token window), the decoding latency per token remains constant regardless of sentence length. The potential issue lies not in per-token latency but in misidentifying mid-utterance pauses as utterance termination, which could prematurely trigger sentence expansion. This can be effectively mitigated by adopting the aforementioned nod-based confirmation strategy for longer expressions.

The discussion on this issue has now been added to the Discussion section:

“Finally, the current end-of-utterance detection strategy—five consecutive blank tokens—ensures zero-delay operation and aligns with the behavioral tendencies of dysarthric stroke patients, who often prefer short, low-effort expressions. For users who wish to communicate longer or pause-rich utterances, the system can readily incorporate an explicit user-controlled cue (e.g., a single nod), which our gesture interface already supports. This modification enables users to indicate expression completion directly, maintaining the system's real-time responsiveness while improving robustness to mid-utterance pauses. Moreover, as the decoding is performed on a fixed sliding window (144 ms per token), the per-token latency and

decoding accuracy remain stable regardless of sentence length.”

Comment 6: The LLM agents (TSA and SEA) are responsible for critical error correction and sentence enrichment, significantly boosting user satisfaction. However, the prompts and few-shot examples used are not provided in the manuscript. It is suggested to include the exact prompt templates and examples as supplementary information. This is crucial for reproducibility and for understanding potential biases the LLM might introduce into the synthesized speech.

Reply: We thank you for this valuable suggestion. We fully agree that reproducibility and transparency of the LLM agents are critical for understanding both performance and potential biases. In the revised manuscript, we have added a new **Note S4** to provide the complete prompt templates and few-shot examples used for both the Token Synthesis Agent (TSA) and the Sentence Expansion Agent (SEA).

Specifically, **Note S4** now presents:

1. The **full prompt text** for the TSA, including the system instruction, task description, and few-shot examples showing input token sequences and corresponding word outputs.
2. The **full prompt text** for the SEA, detailing how emotion labels, time, and environmental context are integrated to produce enriched, contextually coherent sentences.
3. Example pairs of **model input and output** for both agents to illustrate their behavior and correction capability.

These additions ensure full reproducibility of the TSA and SEA configuration, while allowing readers to better understand how prompt design affects synthesis accuracy and bias control. Due to the length of **Note S4**, the full content is not duplicated here but can be found in the Supplementary Information.

Reviewer #3

Comment 1: I recommend the work for publication and only note one thing that wasn't clear. How are pulse signals interpreted to infer emotions? A high pulse could indicate fear, excitement, or even physical exertion.

Reply: We thank you for this insightful question. The emotion inference in our system is not based on absolute pulse amplitude but on the temporal dynamics of the carotid pulse waveform, particularly the R-R interval (RRI) sequence. Emotional changes are known to modulate autonomic nervous activity, which in turn alters the subtle temporal structure of the RRI [1, 2]. Our machine learning model is trained to capture and map these RRI-based temporal representations to corresponding emotional states, thereby learning data-driven associations rather than relying on simple heart rate metrics.

We also fully acknowledge your point regarding the possible confounding effect of physical exertion. This factor was carefully considered in our experimental design. Since our target users are stroke patients with dysarthria, their limited mobility naturally minimizes motion artefacts and exertion-induced variations in pulse signals. This physiological context makes pulse-based emotion decoding feasible using a single sensing modality. To preserve system compactness and comfort, we therefore opted for a single-channel pulse input instead of multimodal fusion. Even with this streamlined configuration, the classifier achieved an accuracy of 83.2%, validating that RRI dynamics contain sufficient discriminative information for reliable emotion decoding within our target population.

[1] Kop, W. J. *et al.* Autonomic nervous system reactivity to positive and negative mood induction: The role of acute psychological responses and frontal electrocortical activity. *Biological Psychology* **86**, 230-238 (2011).

[2] Jauniaux, J. *et al.* Emotion regulation of others' positive and negative emotions is related to distinct patterns of heart rate variability and situational empathy. *PloS One* **15**, e0244427 (2020).

To make it clearer, we also added the following highlighted contents in the first paragraph of the section *Decoding of emotional states*:

“To enrich sentence coherence by providing emotional context, we decode emotional states from carotid pulse signals. Emotional changes modulate autonomic nervous activity, which in turn alters the temporal structure of the R-R interval (RRI) within pulse signals, forming measurable physiological representations of affective states [46]. Our machine learning model establishes a direct mapping between these RRI-based temporal representations and corresponding emotional categories.”

Comment 2: It would be great if the code was available (I may have missed it)

Reply: We thank you for this helpful comment. The code and core dataset supporting this work are publicly available at our GitHub repository: <https://github.com/tcy21414/Intelligent-Throat>.

We have now made this information clearer in the revised manuscript, and it will appear in the Code availability and Data availability statements in the final published version following *Nature Communications* data sharing standards.

NCOMMS-25-62725A R2 Revision Point-by-point Response

Wearable intelligent throat enables natural speech in stroke patients with dysarthria

Reviewer #1

Comment 1: While I appreciate the added analysis on viseme-similar word pairs, the reported performance remains surprisingly high given that distinguishing such pairs is known to be a fundamentally difficult problem in this field. It is somewhat unclear how the system achieves this level of discriminability, especially in a dysarthric patient cohort, where articulatory variability is typically substantial. To better understand and validate this result, I would like to see the raw waveform patterns for these viseme-similar examples and relevance-weighted class activation maps (or an equivalent interpretability method) showing which portions of the raw signal the algorithm model relies on to differentiate these highly similar articulatory gestures.

Reply: We sincerely thank you for this critical observation. We fully agree that discriminating viseme-similar word pairs—such as Book-Look or Metal-Medal—poses a fundamental challenge in speech recognition, particularly in dysarthric populations where articulatory variability is substantial.

To address this concern, we wish to emphasize two key factors that contribute to the observed discriminability in our system:

(1) Building on prior evidence:

Our previous-generation system based on a wearable choker [1] demonstrated that laryngeal surface strain signals can capture sufficient articulatory dynamics to separate viseme-confusable words, achieving ~93% classification accuracy in healthy users. This suggested that even in the absence of vocal output, subtle articulatory differences leave distinguishable signatures at the neck surface.

(2) Advancing to a two-stage decoding architecture:

The Intelligent Throat improves upon this by introducing a token-level decoding framework followed by a large language model (LLM) agent for sentence synthesis. This two-stage process increases robustness for subtle phonemic contrasts by (i) reducing end-to-end dependency on single-word classification errors, and (ii) leveraging majority-voting prompting strategies across multiple token hypotheses to stabilize the decoding [2], which effectively amplifies stable articulatory cues while suppressing noise-induced fluctuations, providing improved separability for phonemically similar tokens. This architecture is particularly advantageous for dysarthric users, as it accommodates intra-subject variability and mitigates the impact of isolated articulation anomalies.

To directly respond to your request for interpretability evidence, we have added **Supplementary Fig. S17**, which visualizes both the raw waveforms and the corresponding 1D Grad-CAM relevance maps [3] for four viseme-similar word pairs tested in our dataset. These visualizations show that the model consistently attends to the core articulatory segments (typically within the 0.8 - 1.8s interval) and allocates high relevance precisely in regions where the paired words diverge (e.g., the beginning part of the dessert-desert pair). The attention maps do not highlight noise artefacts or blank areas without articulation, reinforcing that the network's decisions are grounded in physiologically meaningful deformation patterns.

[1] Tang, C. *et al.* Ultrasensitive textile strain sensors redefine wearable silent speech interfaces with high machine learning efficiency. *npj Flexible Electronics* **8**, 27 (2024).

[2] Ahmed, T. and Devanbu, P. Better patching using llm prompting, via self-consistency. In *2023 38th IEEE/ACM International Conference on Automated Software Engineering (ASE)*, 1742-1746 (2023)

[3] Selvaraju, R. R. *et al.* Grad-cam: Visual explanations from deep networks via gradient-based localization. In *Proceedings of the IEEE International Conference on Computer Vision*, 618-626 (2017).

Fig. S17: Raw waveform patterns and relevance-weighted activation maps (Grad-CAM) for four viseme-similar word pairs.

The following contents have also been added in the Results Section:

“To further evaluate the discriminability of the IT system on visually and articulatorily similar word pairs, we analyzed five viseme-similar pairs (increase/decrease, ship/sheep, book/look, metal/medal, and dessert/desert). The model achieved an average per-word accuracy of 96.3%, with pairwise confusion rates below 8%, indicating that the system can reliably distinguish between look-alike mouth shapes and subtle articulatory gestures. The detailed confusion matrix is shown in Fig. S16. To understand how the system achieves such discriminability, we visualized the raw strain signals and Grad-CAM relevance maps for representative word pairs. As shown in Supplementary Fig. S17, the model consistently focuses on the key articulatory segments where the target words diverge, such as the onset regions in the dessert/desert or book/look pairs. These attention maps confirm that the predictions are driven by meaningful physiological patterns rather than incidental noise or silence segments.”

Comment 2: The explanation of how the 47 words were selected remains insufficient. The authors originally were asked to clarify the sampling protocol—including the use

of a fixed random seed, a reproducible computational pick, and evidence supporting selection-bias mitigation—but the current response only gives a high-level statement that the 350 words were categorized and “randomly” sampled. It is still unclear how the categories were defined, how many words were drawn from each category, whether the selection was computational with a fixed seed or manually chosen after categorization, and how this process can be independently verified. A clear and reproducible description of the actual sampling pipeline is still needed.

Reply: We thank you for this valuable comment and fully agree that a clearer and reproducible description of the corpus-construction procedure is necessary. In the revised manuscript, we have substantially clarified the complete sampling pipeline to remove any ambiguity. Specifically, we now explain that the vocabulary was not selected by manually categorizing words, but instead derived through a fully computational, sentence-first procedure. From the therapist-curated sentence pool used in daily rehabilitation sessions, we randomly selected 20 sentences using a fixed random seed (seed = 42). The final 47 words were then defined as the complete set of unique lexical items that constitute these sampled sentences. This approach ensures unbiased selection, transparent criteria, and full reproducibility, since the corpus can be independently regenerated by applying the same seed to the therapist-provided sentence pool. The **Methods (Study Design section)** has been updated accordingly with the following additions:

“A corpus was developed consisting of 47 Chinese words commonly used by stroke patients in daily communication and 20 sentences constructed from these words (see Supplementary Tables 2 and 3). The corpus originated from the set of sentences routinely used by our collaborating speech-language therapists during daily rehabilitation sessions with dysarthric patients, representing several hundred therapist-curated daily-communication sentences that collectively contain approximately 350 unique words. To ensure a transparent, unbiased, and reproducible sampling protocol, we performed a computational random selection of 20 sentences from this therapist-provided sentence pool using a fixed random seed (seed = 42). The final 47 words represent the complete set of unique vocabulary items that constitute these 20 selected sentences, forming a minimal but comprehensive vocabulary that captures the core communicative intents encountered in real therapy scenarios. This process can be independently reproduced by applying the same seed to the therapist-curated sentence pool and extracting the unique lexical items that compose the sampled sentences.”

Reviewer #2

Comment: The authors have satisfactorily addressed all my comments. I am pleased to recommend acceptance of the manuscript.

Reply: We sincerely appreciate your positive assessment and your constructive feedback throughout the review process, which has greatly strengthened the manuscript.

Reviewer #3

Comment: Thank you to the authors for addressing my prior concerns.

Reply: Thank you for your thoughtful comments and for acknowledging our revisions; your insights were valuable in improving the clarity and quality of the work.